# Time-resolved proteomic profiling reveals compositional and functional transitions across the stress granule life cycle

Shuyao Hu [1] ✉, Yufeng Zhang[1], Qianqian Yi[1], Cuiwei Yang[1], Yanfen Liu [1,2] ✉ & Yun Bai [1,2] ✉

Stress granules (SGs) are dynamic, membrane-less organelles. With their formation and disassembly processes characterized, it remains elusive how compositional transitions are coordinated during prolonged stress to meet changing functional needs. Here, using time-resolved proteomic profiling of the acute to prolonged heat-shock SG life cycle, we identify dynamic SG proteins, further segregated into early and late proteins. Comparison of different groups of SG proteins suggests that their biochemical properties help coordinate SG compositional and functional transitions. In particular, early proteins, with high phase-separation-propensity, drive the rapid formation of the initial SG platform, while late proteins are subsequently recruited as discrete modules to further functionalize SGs. This model, supported by immunoblotting and immunofluorescence imaging, provides a conceptual framework for the compositional transitions throughout the acute to prolonged SG life cycle. Additionally, an early SG constituent, non-muscle myosin II, is shown to promote SG formation by increasing SG fusion, underscoring the strength of this dataset in revealing the complexity of SG regulation.

Stress granules (SGs) are membrane-less organelles that transiently arise in the cytoplasm in response to adverse environmental conditions, such as thermal, oxidative, and osmotic stresses, as well as viral infection[1–3]. SG formation has been associated with global inhibition of translation initiation, which leads to the release of non-translating mRNAs[4] and promotes multivalent interactions among free mRNAs and proteins[5,6]. These events can drive the rapid formation of microscopically well-defined granules via liquid-liquid phase separation (LLPS)[7,8]. Upon stress removal, SGs disassemble promptly through pathways including proteasomal and autophagic degradation[2,9–14].

Throughout their life cycles, SGs are highly dynamic and continuously exchange constituents with surrounding cellular compartments[9,15,16]. Furthermore, SG composition varies with cell type, as well as stress type, intensity, and duration[17–19]. Emerging lines of evidence indicate that their dynamic nature is integral to the physiological functions of SGs. Disruption of SG dynamics by disease-linked

mutations of certain SG constituents, such as TDP-43 and TIA1[20,21], leads to persistent granules, which is associated with pathological inclusions and neurodegenerative diseases, such as amyotrophic lateral sclerosis (ALS) and frontotemporal dementia (FTD)[22].

Although their functions are not fully understood, SGs are thought to regulate mRNA functions during stress[23]. They were initially considered as temporary storage sites for non-translating mRNAs[24,25]. However, recent work indicated that SGs can serve as sites for translation, with translating mRNAs shuttling between SGs and the cytosol[26]. In addition, RNA molecules in SGs can be transferred to other organelles, such as processing bodies and autophagosomes, for further processing[12,27,28]. This RNA triaging function could be particularly relevant during prolonged stress when mRNAs in SGs need to be turned over[29]. Besides, SGs participate in cell-fate specification by coordinating the depletion and release of cytosolic regulatory proteins[30–32]. While they can suppress neuronal cell death as detected

[1]School of Life Science and Technology, ShanghaiTech University, 201210 Shanghai, China. [2]Deceased: Dr. Yun Bai. ✉e-mail: hushy@shanghaitech.edu.cn; liuyf@shanghaitech.edu.cn; ybai@shanghaitech.edu.cn

in degenerative diseases[33], SGs are associated with chemotherapy resistance in cancer cells[34].

Much effort has been focused on understanding the functions and regulation of SGs, including the identification of their regulators and constituents. Genetic screens using siRNA- and CRISPR-Cas9-based approaches led to the identification of a large panel of SG regulators[5,35–38]. In addition, SG constituents have been profiled at proteomic and transcriptomic levels using either purified SGs[24,39] or through proximity labeling with SG proteins, such as G3BP1[40], FMR1[41], and eIF4A1[42]. Using these methods, SG landscapes under several stress contexts, either from wild-type cells or cells harboring disease-linked mutations, have been characterized[43].

Regarding the temporal landscape of SG constituents, previous studies focused on two stages of the SG life cycle, the formation and the disassembly[2,41,42]. However, the stage in between—i.e., the progressive transition of SG composition and function over prolonged stress—remains poorly characterized. Nevertheless, insights about this stage could have physiological and disease relevance, as persistent stress is known to accelerate aging, increase susceptibility to viral infection, and increase risk for neurodegenerative diseases and cancer[18,44–47].

Here, we performed time-resolved proteomic profiling of SGs from cultured human U2OS cells over acute to prolonged heat shock (HS) and a successive recovery period. Altogether, 300 proteins were identified, further clustered into early and late proteins based on their temporal profiles. Each group of SG proteins (early dynamic and late dynamic proteins) share distinct sets of biochemical properties that support their unique SG-enrichment profiles across the acute to prolonged SG life cycle. Together, they coordinate a transition from early SGs that are enriched in RNA-centric functions to late SGs that exhibit enrichment in a diverse range of stress response-related functions, including intracellular protein transport, stress response, and charging of amino acids onto cognate tRNAs, as evidenced by GO enrichment analysis. This model is further supported by evidence from immunoblotting and immunofluorescence imaging of selected SG proteins. Notably, non-muscle myosin II (NM II), a newly identified complex in early SGs, is shown to promote SG fusion and drive SG formation. These findings illustrate the utility of our time-resolved proteomic profiling resource for revealing the diverse layers of SG regulation.

## Results

### Time-resolved proteomic profiling of the acute to prolonged heat-shock stress granule life cycle reveals 300 dynamic SG proteins

Prior to our proteomic profiling, we monitored the acute to prolonged life cycle of heat-shock SGs in U2OS cells with immunofluorescence imaging using the classic SG markers G3BP1 and CAPRIN1. Across a 180 min HS (43 °C) followed by a 10 min recovery (37 °C), microscopically well-defined SGs were observed at the earliest time point (20 min HS), followed by an increase in SG area and a reduction in SG number from 20 min to 120 min HS; a reduction in SG area with no change in SG number from 120 min to 180 min HS; and a moderate reduction in both SG area and number within 10 min recovery (Fig. 1a and Supplementary Fig. 1a).

We used this HS stress time course in subsequent proteomics experiments, sampling at the 7 time points of HS exposure (0, 10, 20, 30, 60, 120, 180 min), as well as 2 time points of recovery (5, 10 min). To comprehensively analyze temporal changes at the proteome-wide level, we have included two different proteins, G3BP1 and CAPRIN1, to enrich for SG proteins from distinct perspectives. From the sampled cells, we prepared SG samples using differential centrifugation plus either endogenous G3BP1 or CAPRIN1 immunoprecipitation protocol[39,48]. Subsequently, peptides were TMT-labeled and analyzed by quantitative LC-MS/MS, with each condition comprising three replicates (Fig. 1b and Supplementary Fig. 1b).

Proteomics analysis of the isolated SG samples identified a total of 442 proteins from the combined G3BP1 and CAPRIN1 pull-down datasets. Among them, 300 proteins showed >1.2-fold changes in comparison to the HS0 control, relative to either the maximum or minimum values observed in the HS10 to Re10 time points, in both the G3BP1 and CAPRIN1 datasets (Supplementary Fig. 1c, d); these were termed as dynamic proteins. To our knowledge, 205 of these 300 proteins have been previously reported as SG constituents (Fig. 1c). Furthermore, within the subset of proteins with fold changes not exceeding >1.2, we identified 85 out of 142 that had been previously reported as SG constituents. We refer to these proteins as invariable proteins, implying their presence within SGs without notable changes in content during prolonged heat shock. Unsupervised hierarchical clustering was next performed[49] to segregate the 300 dynamic proteins. These proteins can be divided into the early and late groups based on sample timing, as indicated by the well-separated peaks in the histogram of component loadings and shown in the heatmap (Fig. 1d, e, Supplementary Data 1). The early group includes 184 early dynamic proteins, or early proteins, whose average profile across the SG life cycle can be fitted to a decreasing sigmoidal curve with an estimated inflection point at 21 min. The late group includes 116 late dynamic proteins, or late proteins, whose average profile data can be fitted to an increasing sigmoidal curve with an estimated inflection point at 30 min (Fig. 1c, f).

### Early vs. late dynamic proteins exhibit distinct phase-separation propensity

We next analyzed and compared the three protein groups (early dynamic, late dynamic, and invariable) to extract features that could participate in the coordination of SG protein recruitment. It is well established that protein-RNA interactions and promiscuous stabilizing protein-protein interactions mediated by intrinsically disordered regions (IDRs) are essential for stress granule formation via phase separation[1,8,50]. We therefore evaluated the phase-separation propensity among the three protein groups by examining three biochemical features: the predicted protein isoelectric point (pI), annotation as an RNA-binding protein (RBP), and the presence/absence of predicted IDRs.

There are significant differences in the distributions of pI among the three protein groups (ANOVA), with the early group featuring the highest pI (median: 8.5) followed by the invariable group (7.2) and the late group (6.7) (Fig. 2a). This suggests that under physiological conditions, early proteins are most inclined to interact with negatively charged nucleic acids. Next, the percentage of annotated RBPs in each group was compared. We found that 65% of the early proteins are annotated as RBPs, which is significantly higher (Chi-square test) than the late proteins (38%) or the invariable proteins (33%) (Fig. 2b). This supports the idea that there is a higher extent of RNA-protein interactions during initial stress-induced SG formation.

We then perform sequence-based IDR prediction for the dynamic and invariable proteins using IUPred2 and Anchor2[51,52], determining an IDR coverage score for each protein (length of predicted IDR regions/total length, per protein). ANOVA indicated a significant difference in IDR coverage among the three groups, with the early group having the highest mean IDR coverage (IUPred2: 29%; Anchor2: 19%), the late group having the lowest (IUPred2: 18%; Anchor2: 11%), and the invariable group in the middle (IUPred2: 23%; Anchor2: 15%) (Fig. 2c). Likewise, PLAAC[53] prediction for prion-like domains (PrLDs), a subset of IDRs known to promote SG formation[54,55], revealed that proteins with PrLDs are significantly more enriched in the early group (21/184), comparing to the late (3/116) and invariable (8/85) groups (Chi-square test; Supplementary Fig. 2a).

These trends specific to the early group are consistent with current knowledge about the typical characteristics of proteins with higher phase-separation propensity[1,8,22,50]. On the contrary, properties

of the late and invariable proteins suggest that a lot of those proteins are not prone to phase separate via protein-RNA or IDR-based protein-protein interactions. These trends suggest a transition in the SG protein recruitment paradigm under prolonged stress.

## Most early dynamic proteins can be recruited into G3BP1-condensates in a stress-independent manner

It is well established that G3BP1 functions as a nucleation protein for SG formation[5] and its overexpression leads to cytoplasmic G3BP1-condensates in the absence of external stress[16,56–58]. We reasoned that

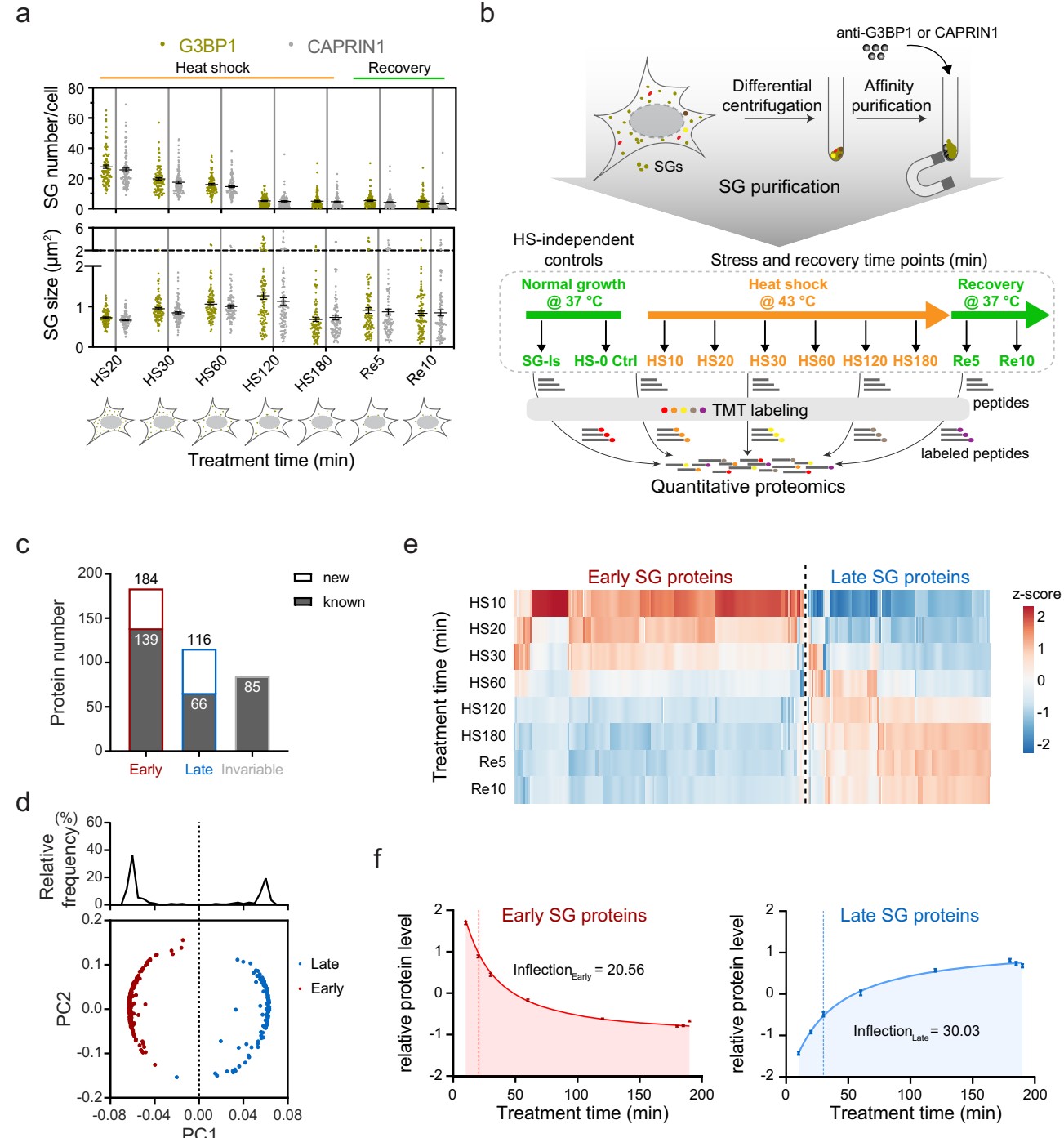

**Fig. 1 | Proteomic profiling of the SG life cycle reveals invariable and dynamics proteins. a** Quantification of SG number and size under HS and recovery conditions; quantification results based on two SG markers (G3BP1 in fern and CAPRIN1 in gray); error bars indicate SEM; n = 100 cells from 3 biological replicates. **b** Schematic overview of the SG life cycle proteomic profiling workflow. **c** Bar graph shows different protein groups. The number within the gray bars indicates the counts of protein previously reported as dynamic proteins. **d** Component loads from the principal component analysis of 300 dynamic proteins; top, histogram of component loadings for the first principal component; bottom, scatter plot of component loadings for the first and second principal component; data points for early and late proteins are labeled in red and blue, respectively. **e** Heatmap of hierarchical clustering of 300 dynamic proteins, highlighting the early and late dynamic proteins. The average intensity of each protein in the CAPRIN1 dataset was scaled, resulting in a z-score. **f** Average profiles for the 184 early proteins (left, red) and the 116 late proteins (right, blue); error bars indicate SEM; fitted curves show average profiles; estimated inflection point for each profile is labeled on the graph. Source data are provided as a Source Data file.

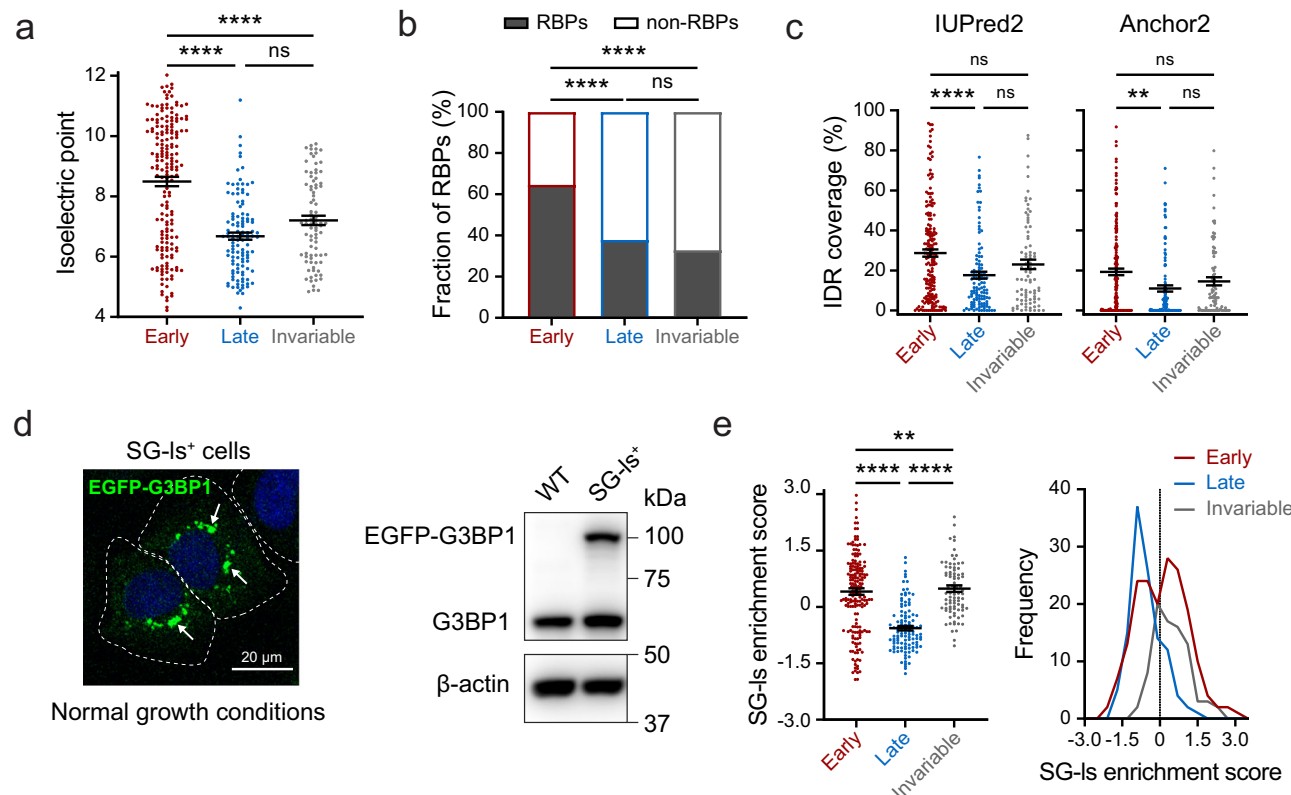

**Fig. 2 | Comparisons among the three protein groups reveal differences in their phase separation propensity. a** Isoelectric points for three protein groups are plotted, with early (red), late (blue) and invariable (gray) proteins; $n = 184$ (early), 116 (late), and 85 (invariable) proteins; error bars indicate SEM; One-Way ANOVA with Tukey's test, exact *P*-values: early vs. late, early vs. invariable, <0.0001 (****); late vs. invariable, 0.0884 (ns). **b** Bar graph showing the percentage of annotated RBPs in each group of proteins, with early (red), late (blue) and invariable (gray) proteins; Two-sided Chi-square test, exact *P*-values: early vs. late, early vs. invariable, <0.0001 (****); late vs. invariable, 0.4661 (ns). **c** IDR coverage scores for three protein groups are plotted, with early (red), late (blue) and invariable (gray) proteins; $n = 184$ (early), 116 (late), and 85 (invariable) proteins; error bars indicate SEM; One-Way ANOVA with Tukey's test, exact *P*-values: early vs. late, IUPred2: <0.0001

(****), Anchor2: 0.0012 (**); early vs. invariable, IUPred2: 0.1246, Anchor2: 0.1605 (ns); late vs. invariable, IUPred2: 0.2079, Anchor2: 0.4093 (ns). **d** Left, a confocal micrograph of SG-like particles (SG-ls), indicated by arrows, in fixed U2OS cells stably overexpressing EGFP-G3BP1; scale bar, 20 μm; representative image from 3 biological replicates was shown; Right, immunoblot of WT and SG-ls positive (SG-ls⁺) U2OS cells for the detection of G3BP1; a representative image from 3 biological replicates is shown. **e** Scatter plot (left) and frequency distribution (right) of SG-ls enrichment scores for three protein groups, with early (red), late (blue) and invariable (gray) proteins; $n = 184$ (early), 116 (late), and 85 (invariable) proteins; error bars indicate SEM; One-Way ANOVA with Tukey's test, exact *P*-values: early vs. late, late vs. invariable, <0.0001 (****); early vs. invariable, 0.0098 (**). Source data are provided as a Source Data file.

an informative control dataset could be produced by overexpressing G3BP1 in unstressed U2OS cells, which should enable the experimental delineation of two types of protein recruitment to G3BP1-condensates: HS-independent recruitment in unstressed cells and HS-dependent recruitment in HS-stressed cells. Pursuing this, we generated a U2OS cell line stably overexpressing EGFP-G3BP1; immunoblotting showed that G3BP1 level in these cells is ~1-fold higher than in wild-type cells, and ~12% of the cells exhibited EGFP-G3BP1-condensates under normal growth conditions (Fig. 2d). We termed these granules as SG-like condensates (SG-ls) and compared their protein composition to isolated SG samples in the G3BP1 pull-down set of TMT-based quantitative MS experiments (Fig. 1b).

To evaluate the enrichment status of each protein in our unstressed SG-ls data, we calculate an SG-ls enrichment score through a scaled fold change calculation, comparing the value of SG-ls to the maximum value observed within the HS10 to Re10 time points. Notably, most early proteins have positive SG-ls enrichment scores (median: 0.13), whereas most late proteins have negative SG-ls enrichment scores (median: −0.57) (Fig. 2e, Supplementary Fig. 2b and Supplementary Data 2). Here, positive scores indicate HS-independent recruitment to G3BP1-condensates, while negative scores suggest HS-dependent recruitment. Interestingly, the early proteins display a bipolar distribution of SG-ls enrichment scores. One cluster has high

SG-ls enrichment scores, while another cluster falls below 0. Notably, 46 out of 67 ribosomal proteins were identified in the cluster below 0. This observation raises the intriguing possibility of ribosomal independence in SG-ls formation (Fig. 2e). Therefore, most early proteins, except ribosomal proteins, can be recruited to G3BP1-condensates in an HS-independent manner. These proteins can interact either directly with G3BP1 or with G3BP1-induced condensate (but not G3BP1). A comparison between SG-ls and HS0 control revealed that a set of proteins are only enriched in SG-ls, suggesting their recruitment to SG-ls through interactions with G3BP1-induced condensates (Supplementary Fig. 2c).

Previously, proximity labeling studies of G3BP1 in the presence and absence of either oxidative stress in HEK293 cells or HS in U2OS cells revealed that many SG proteins form pre-existing interactions with G3BP1 before stress exposure[42,43]. Here our results suggest that G3BP1-condensate-mediated interactions could be an additional mechanism supporting rapid recruitment of early proteins to SG platforms.

**Each group of dynamic proteins makes distinct contributions to SG compositional and functional transitions**

Both protein biochemical properties and experimentally determined SG-ls enrichment scores suggest that, aside from their biological

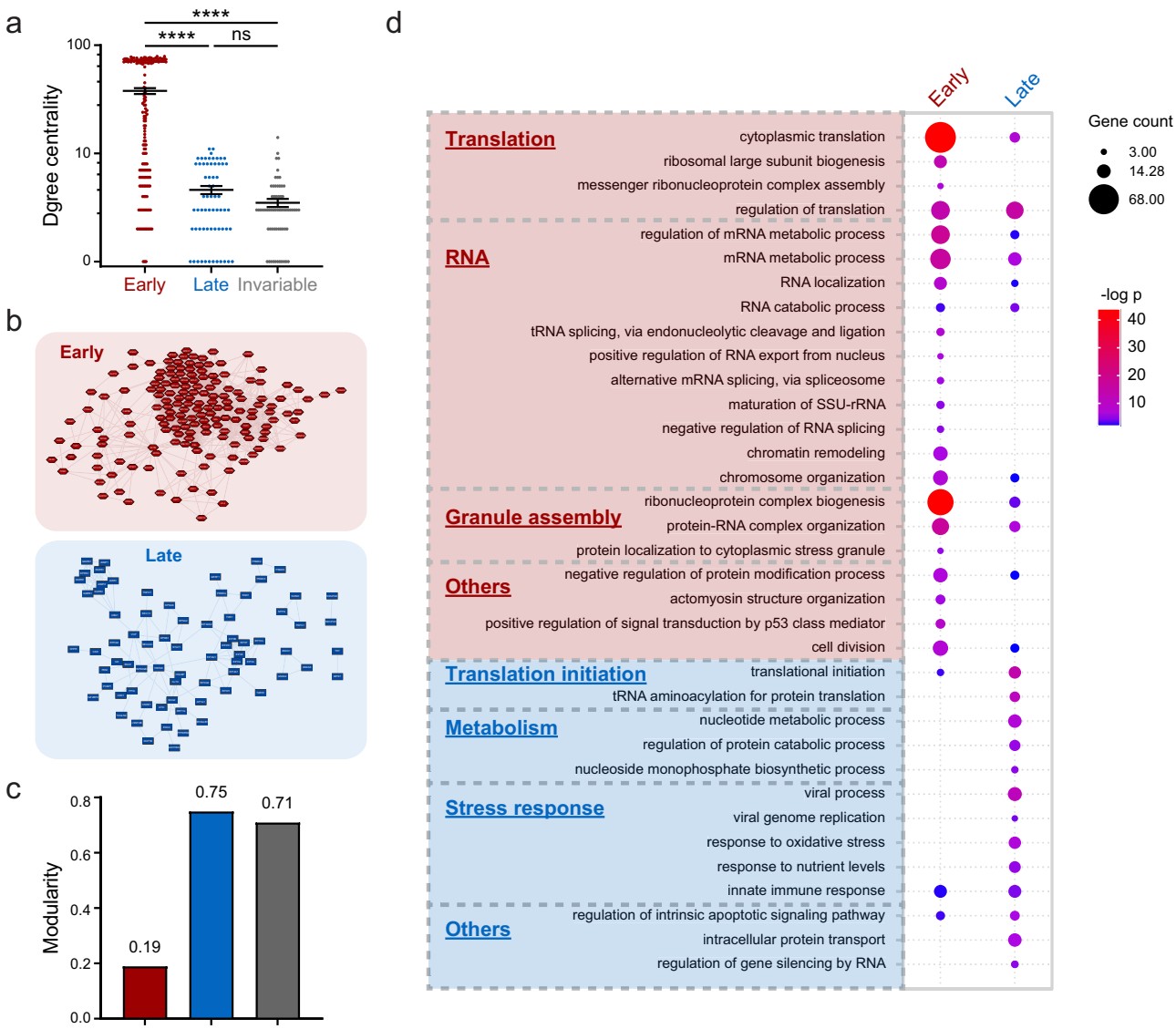

**Fig. 3 | Each group of proteins makes distinct contributions to the organization and function of the SG network. a** Degree centralities for three protein groups are plotted, with early, late and invariable proteins shown in red, blue, and gray, respectively; n = 166, 64, and 63 proteins for early, late and invariable proteins, respectively; error bars indicate SEM; ****P < 0.0001, ns not significant (One-Way ANOVA with Tukey's test, exact P-values: early vs. late, early vs. invariable, <0.0001; late vs. invariable, 0.9624). **b** Top, the subnetwork of early proteins is shown in red; bottom, the subnetwork of late proteins is shown in blue. **c** Bar graph showing the

calculated modularity for subnetworks of the early dynamic (red), late dynamic (blue), and invariable (gray) proteins. **d** Bubble plot of GO enrichment showing biological processes overrepresented by different protein groups; terms particularly enriched by the early (red), late (blue), and invariable (gray) proteins are listed on the left; bubble color represents -log p, p-values are calculated based on the cumulative hypergeometric distribution; bubble size represents the protein number been hit in our list; bubbles are only shown for significantly enriched terms with P < 0.05. Source data are provided as a Source Data file.

functions, early proteins could serve structural roles in rapid SG formation. To further explore this, we retrieved experimentally determined protein-protein interaction (PPI) information for proteins from two databases[59,60] and constructed a PPI network with 364 nodes and 5215 edges (edge information completely independent from our study) (Supplementary Fig. 3a).

Network analysis[61,62] revealed significant differences in topological properties among the three protein groups. In particular, early proteins, with significantly higher degree centralities compared to late and invariable proteins (medians: 37.81, 4.59, and 3.49, respectively; Fig. 3a), might facilitate more transient interactions with other proteins[63]. Additionally, an early-proteins-only subnetwork is densely interconnected, with a network density of 0.23 and an average shortest path length of 2.4. On the contrary, a late-proteins-only subnetwork is more sparse and largely disconnected, with a network density of 0.07

and an average shortest path length of 3.9 (Fig. 3b and Supplementary Fig. 3b, c)[64]. Instead, the late subnetwork forms a highly modular structure with a modularity of 0.75, while the early subnetwork has a lower modularity score of only 0.19 (Fig. 3c)[65,66]. Altogether, topological features suggest early proteins play scaffolding roles in SG formation. In contrast, late proteins form discrete modules, which are recruited to the SG network independent of each other.

A shift in the constituents of SGs can be assumed to reflect changes in their functional capacity(ies). We therefore performed GO analysis[67] and compared the enriched terms among early and late proteins. Note that, consistent with the well-established function of SGs in translation regulation[28], cytoplasmic translation and regulation of translation were enriched in the early protein group, but translation initiation was enriched in the late protein group. This observation might suggest a reinitiation of translation within prolonged heat shock

SGs[26]. SG- and RNP-assembly-related terms, such as protein localization to cytoplasmic stress granule and ribonucleoprotein complex biogenesis were also enriched in the early protein group (Fig. 3d and Supplementary Data 3).

Terms uniquely enriched among the early proteins were largely RNA-centric, including positive regulation of RNA export from the nucleus, mRNA metabolic process, negative regulation of RNA splicing, and maturation of SSU-rRNA, among others. On the contrary, terms enriched in the late protein set covered a diverse range of processes, from intracellular protein transport, regulation of protein catabolic process, viral process, to response to nutrient levels (Fig. 3d and Supplementary Data 3). Together, these data suggest that the compositional transition of SGs supports a switch from RNA-centric processes for early SGs to a more diverse range of stress responses for late SGs.

### Late dynamic proteins are recruited to SGs as discrete functional modules

Late SGs include multi-protein complexes comprising late proteins—i.e., the eukaryotic initiation factor 3 (eIF3), the regulatory complex of the 26S proteasome (PA700), the profilin 1 complex, and the multi-synthetase complex (MSC). Heatmaps of proteomic profiling data revealed that subunits for each of these late complexes share a similar enrichment trend across the SG life cycle, underscoring their recruitment to SGs as structural and functional modules. In particular, all 4 of these modules show delayed enrichment in SGs, leading to peaks at 60 min HS for PA700 and profilin, and 180 min HS for eIF3 and MSC (Supplementary Fig. 4a and Supplementary Data 3).

MSC is composed of 8 aminoacyl-tRNA synthetases and 3 non-synthetase proteins, with 9 out of 11 detected as late proteins in our proteomic profiling. We performed immunoblotting to examine SG-enrichment trends for 5 MSC subunits: IARS, QARS, KARS, EPRS, and RARS. Specifically, their levels were assessed in both total input and SG fractions prepared by differential centrifugation at various HS and recovery time points. Consistent with our proteomic profiling data, all 5 proteins were enriched in SGs, peaking at ~180 min HS. The apparent synchronistic SG-recruitment of these MSC subunits further supports their recruitment as a module (Fig. 4a). Additionally, the enrichment of IARS in SGs during the later time points of heat shock was confirmed by immunofluorescence analysis (Supplementary Fig. 4c, d).

We next performed immunofluorescence imaging on selected late proteins and measured their SG-enrichment levels over the HS SG life cycle, using G3BP1 as an SG marker. Owing to the lack of microscopically distinguishable SGs present at 10 min HS, 20 min HS is the earliest time point in immunofluorescence imaging studies (Supplementary Fig. 1a). The eIF3 complex is a well-documented stress granule constituent, with its subunits used as SG markers in many studies[23,42]. Based on our proteomic profiling results, 6 out of 9 eIF3 subunits are classified as late dynamic proteins. Consistently, we observed eIF3A localization to SGs at the 20 min HS, and its level increased afterward, reaching the peak at 60 min HS (Fig. 4b, c). Beyond bolstering our findings from the proteomic profiling, this observed augmentation of the eIF3A level in late SGs suggests that the potential translation within SGs[26] starts during the later stages of stress response processes.

VCP is an AAA-ATPase family chaperone-like protein[68] that functions in multiple stages of the SG life cycle, including mRNA partitioning during SG formation[69] as well as autophagy- and ubiquitination-dependent SG disassembly/clearance[2,12]. VCP has been previously reported as an SG constituent[10,12,70] and was further classified as a late dynamic protein based on our proteomic profiling results. Immunofluorescence imaging confirmed the SG localization of VCP starting at 120 min HS. VCP level in SGs increases subsequently, peaking at 10 min of recovery (Fig. 4d, e), a finding consistent with the reported function of VCP in SG disassembly[2,10]. Here, our results suggest that the presence, and likely function, of VCP within SGs are only prominent during prolonged HS and recovery. However, it could still function in early stress response either from the cytoplasm or by transiently shuttling to SGs.

The COPII coat complex supports anterograde protein transport and has been reported to mitigate prolonged ER stress in mammalian cells[71]. SG recruitment of this complex was previously observed in HeLa cells under oxidative stress[72]. Our profiling data over the HS time course identified two subunits of COPII coat (SEC24C and SEC13) as late dynamic proteins. Immunofluorescence imaging of SEC24C showed signals in SGs starting at 60 min HS (Fig. 4f, g). These results verify the recruitment of COPII coat to HS SGs and further show that this complex is only involved during prolonged HS.

Altogether, proteomic profiling, immunoblotting, and imaging data showed that late SGs contain functional protein complexes supporting different cellular processes, such as translation control (eIF3 and MSC), protein turn-over (PA700 and VCP), and anterograde protein transport (SEC24C). Each of these complexes shows distinct SG-enrichment kinetics, implying their recruitment as discrete modules, with each module executing additional functions in SGs.

### RPS6 and some other early dynamic proteins dissociate from SGs prior to their disassembly

Early SGs are particularly enriched in proteins with RNA-related complexes, such as components of the 40S ribosomal subunit, the C complex of spliceosome, and the Large Drosha complex. These proteins rapidly accumulate in SGs, leading to the highest SG levels at the earliest sampled time point (10 min HS). SG levels of the 40S subunit showed a rapid decline at 20 min HS, while SG levels of the Large Drosha complex and the C complex of spliceosome started to decline at 60 min HS (Supplementary Fig. 4b and Supplementary Data 3). These findings suggest that, after the formation of SGs, some early proteins could relocate to the cytoplasm at some point prior to SG disassembly, exhibiting different dissociation kinetics.

We next performed immunofluorescence imaging on selected early proteins and measured their SG-enrichment levels over the SG life cycle, using G3BP1 as an SG marker. The 40S subunit is one of the best-studied SG constituents[23,42] and has been proposed to be recruited to SGs together with non-translating mRNAs early during SG formation[73]. However, it is unclear whether the 40S subunit remains associated with SGs throughout the prolonged SG life cycle. Our proteomic profiling results indicated that several components of the 40S subunit are early dynamic proteins. Using ribosomal protein RPS6 as a 40S marker, immunofluorescence imaging revealed the highest SG enrichment at 20 min HS, followed by a rapid decline. At 60 min HS, RPS6 was no longer colocalized with G3BP1; rather, its signal became diffuse in the cytoplasm at this time point and afterward (Fig. 5a, b). Notably, although SG recruitment of translation initiation factors is usually considered to occur concurrently with SG recruitment of the 40S subunit (both triggered by global inhibition of translation initiation)[23,74], our data clearly showed different SG recruitment trends for the 40S subunit vs. eIF3.

Our proteomic profiling also identified poly(ADP-ribose) polymerase 1 (PARP1), the most abundant and active member of the PARP protein family and a major target for a class of anti-cancer therapies, as an early dynamic protein. Poly(ADP-ribosylation) (PARylation) has been reported to regulate the formation of physiological and pathological SGs[75,76]. Several PARPs have been reported as constituents of oxidative stress-induced SGs[77], and PARP1 has also been reported to be increased in a 2x IDR mutant G3BP1 SGs[5]. PARP1 localizes and functions mainly in the nucleus[78,79] but also in the cytoplasm[80]. Immunofluorescence imaging shows that, although predominantly localized in the nucleus over the HS cycle, a minor fraction of PARP1 was enriched in SGs starting at 20 min HS. Its signal remained enriched till 60 min HS before becoming diffuse at 120 min HS (Fig. 5c, d). Consistent with our proteomic results, this observed

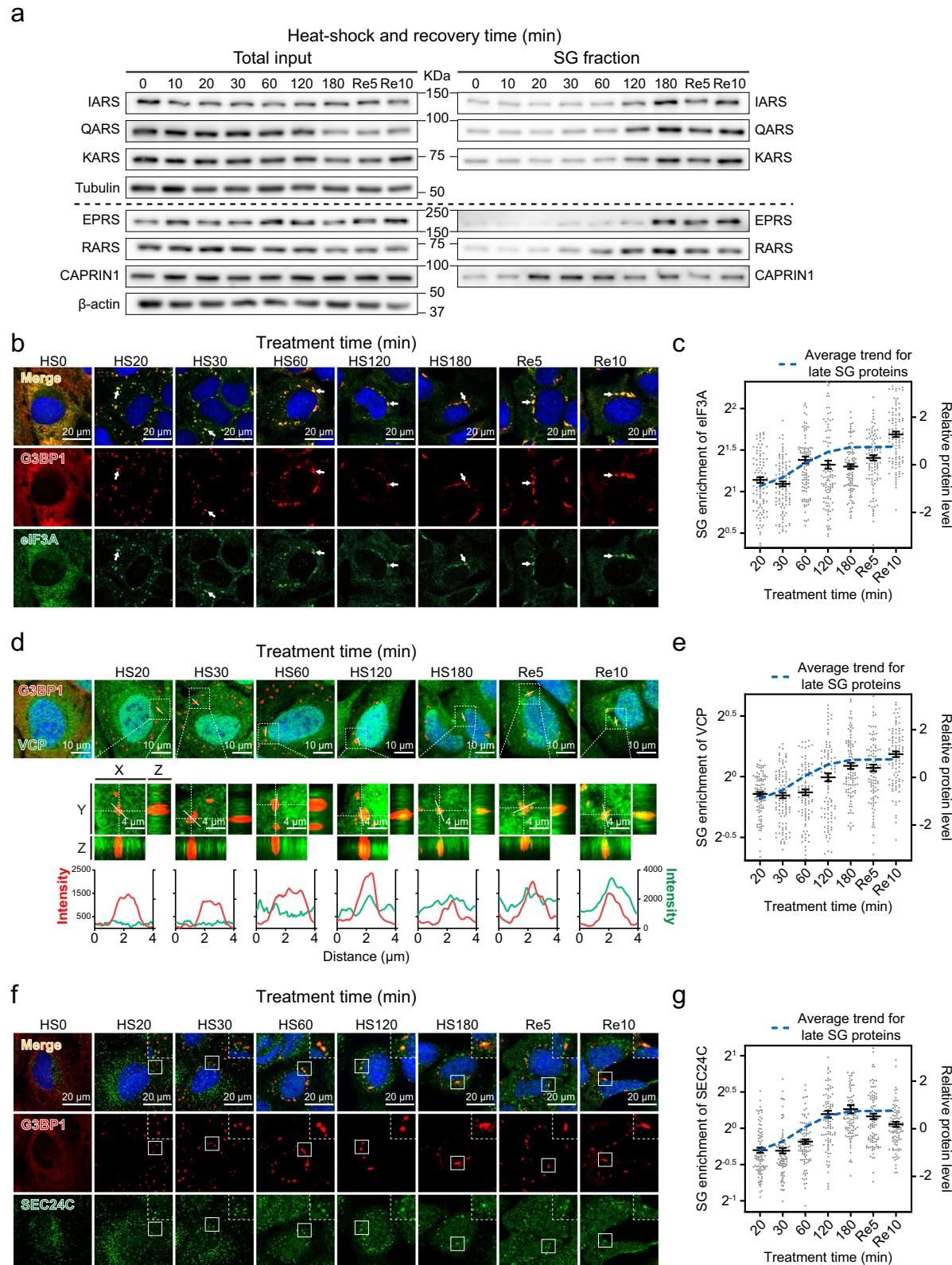

early SG-enrichment pattern suggests that PARP1 functions within SGs during early HS stress response.

TDP-43, a well-established SG constituent[S1], was also identified as an early dynamic protein in our proteomic profiling. Under normal physiological conditions, TDP-43 is primarily localized in the cell nucleus, where it is involved in RNA processing and splicing. However, under cellular stress, TDP-43 can translocate from the nucleus to the

cytoplasm and become sequestered within SGs[S1]. In specific neuro-degenerative diseases, such as ALS and FTD, TDP-43 is known to accumulate abnormally in the cytosol, forming pathological aggregates that disrupt normal cellular functions[S2]. Our proteomic profiling identified TDP-43 as an early dynamic protein. This observation was corroborated by immunofluorescence imaging, which showed the localization of TDP-43 within SGs, starting at 20 min HS and peaking in

**Fig. 4 | Delayed and modular SG recruitment of late dynamic proteins.**
**a** Immunoblots of the total input and the SG fraction samples collected at indicated HS and recovery time points for the detection of IARS, QARS, KARS, EPRS, and RARS; CAPRIN1 is a SG constituent control in the SG fraction; representative blots from 3 biological replicates are shown. **b**, **d**, **f** Confocal micrographs of U2OS cells fixed at the indicated time points during HS and recovery; immunofluorescence of G3BP1 and eIF3A (**b**), VCP (**d**), or SEC24C (**f**) are shown, SGs were indicated by G3BP1, presented in red, other proteins were presented in green; arrows (**b**) or insets (**f**) highlight colocalization of two proteins, scale bars, 20 μm; top of (**e**), z-stack projections, scale bars, 10 μm; middle of (**e**), magnified orthogonal sectioning view of regions in boxes, scale bars, 4 μm; bottom of (**e**): line profiles of G3BP1 and VCP fluorescence intensity along the white solid lines; representative images from 3 biological replicates are shown. **c**, **e**, **g** Enrichment of eIF3A (**c**), VCP (**e**), and SEC24C (**g**) fluorescence intensity in G3BP1-positive granules over that in the whole cell, calculated per cell; error bars indicate SEM; the fitted trend of late proteins based on proteomic profiling results (Fig. 1f) is shown in blue as a reference; n = 102 (**c**), 100 (**e**, **g**) cells for every time point, from 3 biological replicates. Source data are provided as a Source Data file.

enrichment at 30 min HS. Its signal becomes diffuse at 120 min HS (Fig. 5e, f). Consistent with our proteomic results, this observed early SG-enrichment pattern suggests that TDP-43 functions within SGs during the early stages of the heat shock stress response. In addition, these results underscore the capacity of a SG life cycle profiling approach to identify proteins with only a small cytoplasmic fraction transiently recruited to SGs.

Notably, using RPS6, PARP1, and TDP-43 as examples, our results reveal that some SG proteins can dissociate from granules well ahead of the disassembly of entire SGs upon stress relief. This highlights underappreciated mechanisms that regulate the selective dissociation of protein constituents from SGs.

### Subunits of the non-muscle myosin II complex are early proteins that promote stress granule formation

The formation of SGs requires physical interactions between SG constituents, as well as the gathering of SG constituents through both passive diffusion and active transport. The microtubule network and its associated motors have been reported to be involved in the latter process[83,84]. Although the actin filament network has the potential for driving the coalescence of cytoplasmic condensates, its involvement in SG formation is controversial[83,85,86]. Here, our proteomic profiling reveals non-muscle myosin II complex (NM II), a molecular motor with actin cross-linking and contractile properties[87], as early proteins. This complex comprises four early dynamic proteins, MYH9, MYH10, MYL12B, and MYL6, all sharing a similar SG-enrichment trend over the examined SG life cycle (Fig. 6a and Supplementary Fig. 5a), with a peak at 20 min HS followed by a rapid decline. Based on this profile, we hypothesized that active processes involving NM II could affect the formation of SGs.

To test this hypothesis, we pharmacologically inhibited the ATPase activity of the NM II heavy chain using a well-characterized and widely-used NM II inhibitor, blebbistatin (blebb)[88]. In light of our data showing that the SG levels of all NM II subunits peak at 20 min HS, we performed immunofluorescence imaging on both blebb- and vehicle-treated U2OS cells at this HS time point, using G3BP1 as a marker for SGs. A moderate (24%) but significant reduction of SG number was observed in blebb-treated cells (Fig. 6b), suggesting that the NM II complex may promote SG formation.

To determine if this apparent capacity of NM II to promote SG formation is specific to HS, we repeated the NM II-inhibition experiments but replaced HS with NaCl-induced osmotic stress. Live cell imaging of U2OS cells over 60 min of osmotic stress (using EGFP-G3BP1 as a marker for SGs) revealed that blebb-mediated NM II-inhibition caused a significant delay in SG formation and a 42% reduction in SG number at 60 min of stress (Fig. 6c, d). Similarly, when U2OS cells were stressed with vinorelbine, a widely-used microtubule-destabilizing chemotherapy drug that induces SG formation[89], NM II inhibition hampered SG formation even more severely, leading to a 73% reduction in SG number at 120 min of stress (Supplementary Fig. 5b, c).

Because HS plus NM II inhibition leads to severe phototoxicity in live cell imaging, results from osmotic stress were used for further analysis. Imaging analysis of blebb-treated cells revealed conspicuous trends for SG size and mobility. Specifically, whereas the SG size for vehicle-treated cells showed an obvious sigmoidal-like increase with stress time and plateaued at ~30 min of stress, there was only a marginal increase in the SG size over the entire 60 min imaging window in blebb-treated cells (Fig. 6e). We also noticed that osmotic-stress-induced SGs in blebb-treated cells were less mobile (Fig. 6c and Supplementary Movie 1). Particle-tracking[90] confirmed that blebb-treatment causes a significant reduction in the mobility of SGs in osmotic-stressed cells (Fig. 6f).

We next speculated that NM II-mediated alteration of particle size and mobility may affect SG formation by promoting the coalescence of granules[91]. To pursue this, 50 regions of interest (ROIs) were randomly selected from live cell imaging of blebb- and vehicle-treated cells, and SG fusion events in those ROIs were manually inspected. The result showed that SG fusion happens much more frequently in vehicle-treated cells (46/50) than in blebb-treated cells (14/50; Chi-square test; Supplementary Fig. 5d, e).

To further investigate the role of the NM II complex in SG formation, we employed a gain-of-function strategy. Thr18/Ser19 of MYL9 are well-established phosphorylation sites that activate the NM II complex[87]. Therefore, we investigated the role of the NM II complex in SG assembly by activating its regulatory subunit, MYL9. To achieve this, we transfected cells with a phosphomimetic MYL9 variant containing a T18D/S19D mutation (MYL9^T18D/S19D), which is known to enhance myosin functions[92]. Subsequent live cell imaging revealed that MYL9^T18D/S19D significantly accelerated SG formation compared to the wild-type MYL9. These results strongly support the crucial role of the NM II complex in SG formation, a finding that resonates with the inhibitory effects observed with Blebb (Fig. 6g and Supplementary Movie 2).

Collectively, proteomic profiling suggests NM II as constituents of early SGs, and transient pharmacological inhibition experiments reveal that this complex can promote SG formation under various stress contexts. Detailed analysis of osmotic-stress-induced SG particles shows that active NM II enhances the mobility of SGs and promotes the coalescence of smaller granules into larger SGs.

## Discussion

In this study, we profiled the proteomic landscape of SGs over the acute to prolonged SG life cycle, with a special focus on their compositional transitions under prolonged HS. It has been well-established that stress granules assemble via LLPS facilitated by multivalent interactions[1,5,6,8,93]. However, the driving force behind progressive compositional and functional changes of SGs under persistent stress remains largely unknown. SG life cycle profiling enables us to group SG proteins, based on differences in their temporal behaviors, into the early dynamic SG proteins and the late dynamic proteins. In-depth comparison of these two protein groups identified informative differences among them. In particular, early dynamic proteins, with an average pI of 8.5, 65% of annotated RBPs, high IDR and PrLD scores, and a median of 37.81 degree centralities in the SG PPI network, are primed for phase separation based on RNA-protein and protein-protein interactions. In contrast, the late dynamic proteins, with significantly lower average pI, percentage of annotated RBPs, IDR and PrLD scores, and less direct interaction partners in the SG network, are not as phase-separation-prone as early proteins. Instead, late proteins

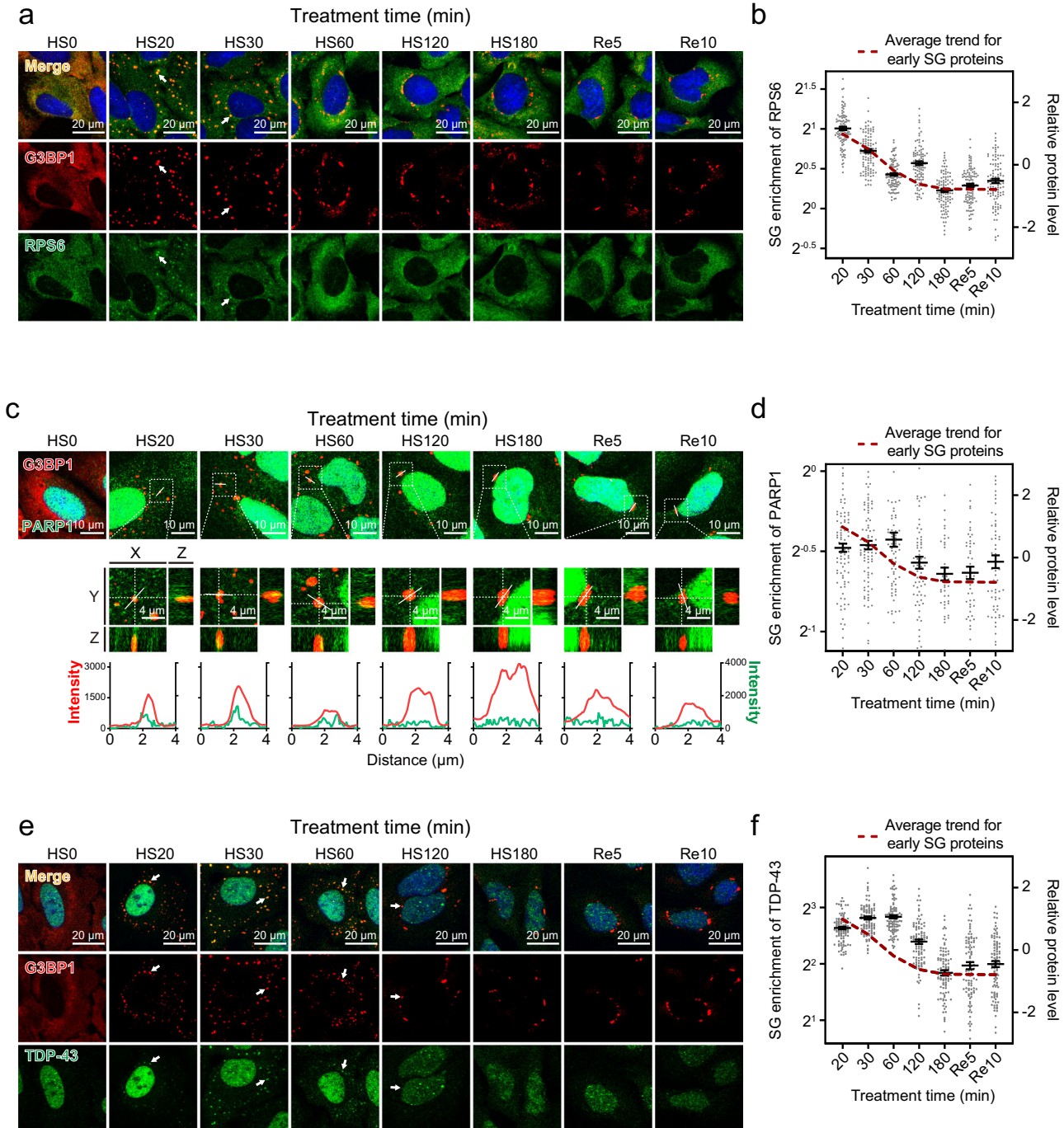

**Fig. 5 | Early dynamic proteins dissociate from SGs during prolonged HS.**
**a**, **c**, **e** Confocal micrographs of U2OS cells fixed at the indicated time points during HS and recovery; immunofluorescence of G3BP1 and RPS6 (**a**), PARP1 (**c**), or TDP-43 (**e**) are shown, SGs were indicated by G3BP1, presented in red, other proteins were presented in green; arrows (**a**, **e**) highlight colocalization of two proteins, scale bars, 20 μm; top of (**c**), z-stack projections, scale bars, 10 μm; middle of (**c**), magnified orthogonal sectioning view of regions in boxes, scale bars, 4 μm; bottom of (**c**): line profiles of G3BP1 and PARP1 fluorescence intensity along the white solid lines; representative images from 3 biological replicates are shown. **b**, **d**, **f** Enrichment of RPS6 (**b**), PARP1 (**d**), and TDP-43 (**f**) fluorescence intensity in G3BP1-positive granules over that in the whole cell, calculated per cell; error bars indicate SEM; the fitted trend of early proteins based on proteomic profiling results (Fig. 1f) is shown in red as a reference; $n = 102$ (**b**, **f**), 114 (**d**) cells for every time point, from 3 biological replicates. Source data are provided as a Source Data file.

form discrete modules via specific protein-protein interactions (Fig. 3b). Proteomic analysis of SG-like particles (SG-ls), the cellular condensates in unstressed U2OS cells nucleated by the overexpression of G3BP1, further demonstrated the difference between the early and late dynamic proteins in their propensity for being enriched in G3BP1-induced condensates. While most early proteins are enriched in SG-ls, only a small fraction of late proteins show enrichment in those stress-independent condensates (Fig. 2e and Supplementary Fig. 2b).

Based on these results, we propose the following model for the dynamic transition of SG compositions and functions. Upon stress exposure, early proteins, which are highly capable of multivalent interactions, can rapidly phase separate into early SGs. GO process enrichment supported by early proteins, such as regulation of translation, RNA stability, and RNA localization (Fig. 3d), are highly RNA-centric, indicating that early SGs could serve as a hub for RNA-related cellular responses[94]. As stress continues, late proteins, which

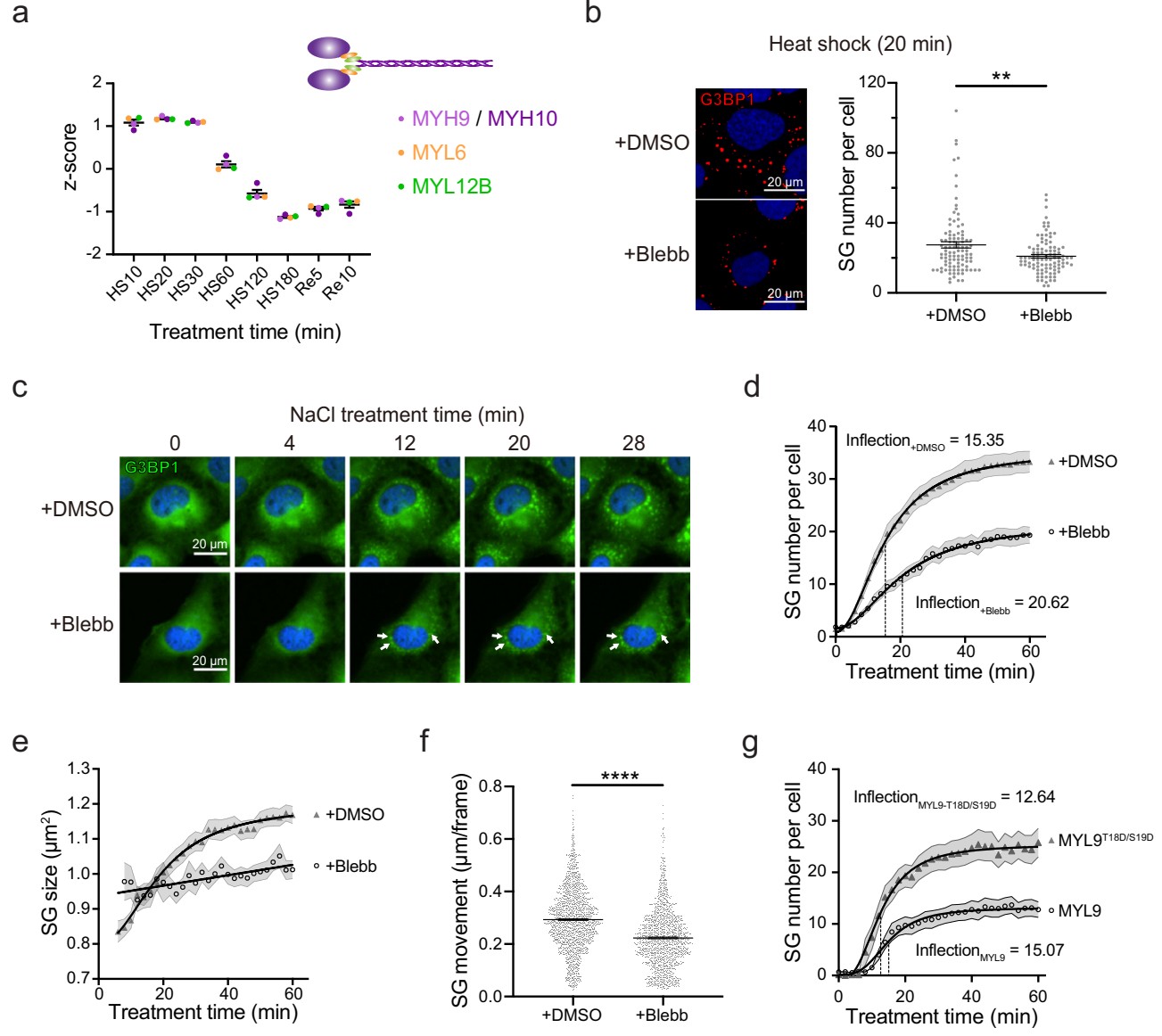

**Fig. 6 | NM II is an early dynamic SG constituent that promotes SG formation.**
**a** The relative protein levels (shown as *z*-scores) for the four subunits of the NM II complex, MYH9 (light purple), MYH10 (dark purple), MYL6 (orange), and MYL12B (green) at different HS and recovery time points; error bars indicate SEM; *n* = 4 proteins; a cartoon model of the NM II complex is shown on the top right. **b** Left: confocal micrographs of U2OS cells pre-treated with DMSO or blebbistatin; G3BP1 is shown in red; scale bars, 20 μm; representative images from 3 biological replicates are shown. Right: quantification of SG number per cell with *n* = 100 cells from 3 biological replicates for each condition; error bars indicate SEM; Two-sided Welch's t-test, *P*-value = 0.0019. **c** Snapshots from live cell imaging of U2OS cells stably expressing EGFP-G3BP1 pre-treated with DMSO or (S)−4′-nitro-blebbistatin; scale bars, 20 μm; arrows show granules of low mobility; representative images

from 3 biological replicates are shown. **d** Quantification of SG number per cell under osmotic stress based on live cell imaging (**c**); *n* = 69 cells from 3 biological replicates; shadows indicate SEM; estimated inflection points are shown as dash lines. **e** Quantification of SG size under osmotic stress time based on live cell imaging (**c**); *n* = 69 cells from 3 biological replicates; shadows indicate SEM. **f** Mobility of osmotic-stress-induced SGs based on live cell imaging (**c**); *n* = 1725 (DMSO) and 1563 ((S)−4′-nitro-blebbistatin) detected movement; error bars indicate SEM. **g** Quantification of SG number per cell under osmotic stress based on a live cell imaging of EGFP-G3BP1 U2OS cells transiently overexpressing MYL9 or MYL9[T18D/S19D]; *n* = 30 cells from 3 biological replicates; shadows indicate SEM; estimated inflection points are shown as dash lines. Source data are provided as a Source Data file.

are not capable of forming condensates on their own, are recruited to already-established SG platforms as discrete structural and functional modules. Delayed SG recruitment of these modules, which include the eukaryotic initiation factor 3 (eIF3), the PA700 complex, and the multisynthetase complex, execute a diverse range of stress-related functions in late SGs[71,95,96]. Notably, SG recruitment of these modules seems to be independent of each other, supported by the distinct recruitment kinetics of different late SG complexes (Supplementary Fig. 4b). Therefore, different combinations of these

modules (in an on-demand manner) can potentially contribute to SG compositional and functional diversity as detected in distinct cellular and stress contexts[43,97] (Fig. 7). Nevertheless, further studies are required to elucidate and compare the SG-recruitment kinetics of these late modules−i.e., the proteasome and VCP−under different stress conditions and in the context of neurodegenerative disease-linked mutations.

SG formation is an ATP-dependent process involving both passive diffusion and active intracellular transport[39,84]. Several lines of

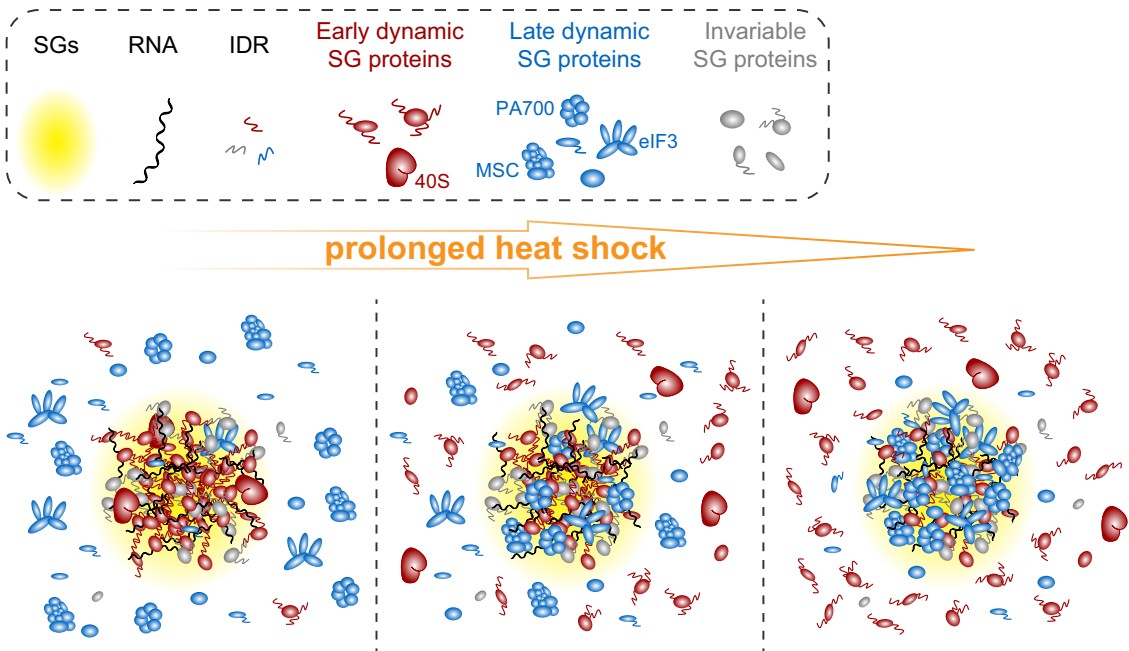

**Fig. 7 | A model for SG compositional transitions under prolonged HS.** SGs are highlighted in yellow. RNA molecules are shown in black; Early dynamic, late dynamic, and invariable proteins are shown in red, blue, and gray, respectively.

evidence support the engagement of microtubules and associated molecular motors, such as kinesin and dynein, in SG dynamics[98,99]. However, the role of actin filaments in SG dynamics remains controversial. While their disruption was shown not to affect SG mobility in oxidative-stressed mammalian cells[83,100], actin filaments were reported to contribute to SG movement in plant cells[85,86]. Our present study extends the current understanding of how intracellular transport machinery affects SGs by introducing the NM II complex, a ubiquitous actin-associated molecular motor, as an early dynamic SG constituent, and demonstrating that the motor activity of NM II contributes to SG formation under various stress contexts. Transient pharmacological inhibition of NM II leads to a delay in SG formation and a decrease in SG number. Particle-tracking of microscopically well-defined G3BP1-positive foci suggests that the NM II complex enhances the mobility of those foci and promotes the fusion of small foci into larger SGs. This resembles the coalescence of SGs via microtubule-dependent transport during SG formation[98] and is consistent with a previous report showing that NM II motors generate active forces to drive the diffusive-like motion of vesicles and organelles[101]. Nevertheless, further studies are necessary to investigate whether actin filaments function together with the NM II complex in promoting the formation of SGs.

Exploring SG constituents through the pull-down of specific SG markers has inherent limitations, such as restricted coverage of the SG proteomics, a potential bias toward G3BP1-interacting proteins, and possible disruption of SG integrity. Nevertheless, in our study, the temporal proteomics analysis of SGs under prolonged heat shock-induced conditions was conducted using endogenously expressed G3BP1. This approach eliminates the potential artifacts associated with the use of overexpressed genes during SG purification. Importantly, we performed endogenous SG purification using two SG markers (G3BP1 and CAPRIN1) separately and only retained those co-identified in both pull-down experiments, which largely enhanced the specificity of our reported SG components.

In summary, our proteomic profiling of the prolonged SG life cycle established a conceptual framework to explain the compositional and functional transitions of SGs. By introducing a temporal vector to the SG proteome, this resource allows finer-scale elucidation of sub-processes in the SG life cycle, as shown by the identification of the NM II complex as a constituent of early SGs and the demonstration of its function in promoting SG formation.

## Methods

### Cell lines

Two U2OS cell lines stably expressing EGFP-G3BP1 were generated by lentiviral infection followed by a selection using 5 µg/ml puromycin. EGFP-positive cells were further sorted by fluorescence-activated cell sorting (FACS) to generate single-cell colonies, and EGFP-G3BP1 expression was validated by immunoblotting and immunofluorescence imaging. One of the generated cell lines harbors EGFP-G3BP1-positive granules under normal growth conditions and was maintained as the SG-ls positive cell line. In another cell line, EGFP-G3BP1 is diffuse under normal growth conditions but forms cytosolic granules upon stress. This cell line was maintained for live cell imaging studies. U2OS cells were from females.

### Cell culture

U2OS cells (TCHu 88, Cell Bank/Stem Cell Bank, National Collection of Authenticated Cell Cultures) were cultured at 37 °C with 5% $CO_2$ in Dulbecco's Modified Eagle Medium (DMEM) + GlutaMAX (Gibco, 10566024) supplemented with 100 U/mL penicillin, 100 µg/mL streptomycin, and 10% fetal bovine serum.

### Construction of plasmids

EGFP-G3BP1 plasmid was constructed by restriction enzyme digestion and ligation into the pCDH vector. EGFP-G3BP1 fragment was generated using the primers listed here: forward: GTGCTCTAGAATGGTG AGCAAGGGCGAGGAG; reverse: GCATGGATCCTC ACTGCCGTGGCG-CAAGCC. MYL9 and MYL9[T18D/S19D] plasmids were constructed by PCR and recombination into pCDNA3, with a mCherry tag being ligated at the C terminus of MYL9 and MYL9[T18D/S19D]. MYL9 and MYL9[T18D/S19D] fragments were generated using the primers listed here: forward: CACTATAGGGAGACCCAAGCTTATGTCCAGCAAGCGGG; reverse: ATC CGAGCTCGGTACCAAGCTGTCGTCTTTATCCTTGGCGC. pCDNA3 was linearized using the primers listed here: forward: CTTGGTACC-GAGCTCGGA; reverse: CTTGGGTCTCCCTATAGTGAGT.

## Heat shock and recovery, NaCl, vinorelbine, and blebbistatin treatment

U2OS cells were cultured to 80–90% confluency before stress. For heat shock, stress granules were induced by incubation at 43 °C with 5% $CO_2$ for an indicated period of time. For recovery, heat-shocked cells were incubated at 37 °C with 5% $CO_2$ for either 5 or 10 min. For osmotic stress, stress granules were induced with 0.2 M NaCl. For stress with vinorelbine (APExBIO, N2250), stress granules were induced with 100 μM vinorelbine. Pharmacological inhibition of NM II activity was achieved by pre-treating cells for 3 h before stress, with either 100 μM (-)-blebbistatin (Beyotime, SF9087-5mg) for immunofluorescence imaging experiments or 100 μM (S)-4′-nitro-blebbistatin (Cayman Chemical, 24171) for live cell imaging experiments.

## SG sample collection and purification

For each sample, wild-type U2OS cells were grown to 80–90% confluency in five 150 mm tissue culture dishes before stress. After treatment, cells were washed with and scraped in PBS, and collected by 1500×*g* centrifugation for 1 min. Cell pellets were flash-frozen in liquid nitrogen and stored at −80 °C.

Stress granules were purified as previously reported[39,48]. To obtain the SG fraction, frozen cell pellets were thawed in SG Lysis Buffer (50 mM Tris-HCl pH 7.4, 100 mM KOAc, 2 mM Mg(OAc)$_2$, 50 μg/mL Heparin, 0.5% NP-40, 1:5000 antifoam B, 0.5 mM DTT, 1 U/μL RNasin® Plus RNase Inhibitor (Promega, N2615), and protease inhibitor (APExBIO, K1010; added immediately before use)) on ice and lysed by passing through a 25G needle for 7 times on ice. An aliquot of the lysate was saved for the preparation of a total input protein sample. The rest of the lysate was centrifuged at 1000 × *g* for 5 min at 4 °C. The supernatant was carefully collected and further centrifuged at 18,000×*g* for 20 min at 4 °C. The resulting pellet was washed and resuspended in SG Lysis Buffer, and centrifuged at 850 × *g* for 2 min at 4 °C. The resulting supernatant is the SG fraction (S850).

For further affinity purification of SGs, S850 was pre-cleared by incubation with equilibrated protein A/G magnetic beads (Bimake, B23202) for 30 min at 4 °C. After beads removal with a magnet, supernatant was transferred to a new tube, mixed with either G3BP1 or CAPRIN1 antibody-crosslinked protein A/G magnetic beads prepared immediately before use, and incubated for 3 h at 4 °C. Beads were washed for 5 min at 4 °C with SG Lysis Buffer, repeated 3 times, then washed once with Wash Buffer I (SG lysis Buffer + 2 M urea) for 2 min at 4 °C, washed once with Wash Buffer II (similar to SG lysis Buffer but changing the KOAc concentration to 300 mM instead of 100 mM) for 2 min at 4 °C, and finally washed twice with SG Lysis Buffer, for 5 min at 4 °C each time. SG-conjugated beads were resuspended with Reducing Buffer (50 mM $NH_4HCO_3$, and 8 M Urea), supplemented with 20 mM DTT and incubated at 37 °C for 1 h with agitation. Next, iodoacetamide (IAM) was added to a final concentration of 40 mM, and the beads were incubated at room temperature for 1 h in the dark with agitation. The reaction was quenched with the addition of DTT to a final concentration of 10 mM. SG proteins were subsequently isolated using TRIzol Reagent (Invitrogen, 15596018) according to the manufacturer's protocol and precipitated in 6 volumes of −20 °C pre-chilled acetone.

## TMT-based quantitative proteomics

Acetone precipitated proteins were re-dissolved with 50 mM TEAB and digested with Trypsin Protease (Thermo Scientific, 90057) at 37 °C overnight. Peptides were TMT-labeled (Thermo Scientific, 90309) according to the manufacturer's protocol. TMT-labeled peptides were combined and dried in a speed vac, desalted using MonoSpin™ C18 (GL Sciences, 5010-21700) according to the manufacturer's protocol, and dried in a speed vac. Peptides were separated and analyzed on an EASY-nLC 1200 system coupled to a Q Exactive HF (Thermo Scientific). About 1 μg of peptides were separated in a homemade column (75 μm × 15 cm) packed with C18 AQ (1.9 μm, ReproSil-Pur 120 dr.maisch

Germany) at a flow rate of 300 nL/min. Mobile phase A (0.1% formic acid in 2% ACN) and mobile phase B (0.1% formic acid in 98% ACN) were used to establish a 60 min gradient composed of 42 min of 30% B, 10 min of 50% B, 2 min of 90% B, 6 min 90% B. Peptides were then ionized by electrospray at 2.1 kV. A full MS spectrum (350–1400 m/z range) was acquired at a resolution of 60,000 at m/z 200 and a maximum ion accumulation time of 50 ms. Dynamic exclusion was set to 30 s. Resolution for HCD MS/MS spectra was set to 30,000 at m/z 200. The AGC setting of MS1 and MS2 were set at 3E6 and 1E5, respectively. The 20 most intense ions above a threshold of $1 \times 10^4$ counts were selected for fragmentation by HCD with a maximum ion accumulation time of 60 ms. MS2 isolation width of 2 m/z units was used. Single and unassigned charged ions were excluded from MS/MS. For HCD, normalized collision energy was set to 32%.

## TMT-quantitative mass spectrometry was performed in triplicate for both G3BP1 and CAPRIN pull-down experiments

**Immunofluorescence microscopy.** U2OS cells were plated onto poly-lysine-coated coverslips and cultured for 2 days. After stress treatment, cells were fixed with 4% paraformaldehyde (PFA) in PBS for 15 min at room temperature and washed with PBS three times. Cells were permeabilized and blocked using QuickBlock™ Blocking Buffer for Immunol Staining (Beyotime, P0260) and incubated with primary antibody in QuickBlock™ Primary Antibody Dilution Buffer for Immunol Staining (Beyotime, P0256) for overnight at 4 °C. Cells were next washed with PBS for 3 times and incubated with secondary antibody in QuickBlock™ Secondary Antibody Dilution Buffer for Immunol Staining (Beyotime, P0265) for 2 h at room temperature. Coverslips were finally washed with PBS 3 times and mounted on a slide with mounting medium (BBI Lifescience, E675003). Images were collected using a Leica SP8 STED 3X inverted microscope with LAS X (Leica, 3.5.7.23225) software. Primary antibodies used for IF: mouse anti-G3BP1 monoclonal [2F3] antibody (Abcam, ab56574, 1:500), rabbit anti-G3BP1 polyclonal antibody (Proteintech, 13057-2-AP, 1:500), rabbit anti-CAPRIN1 polyclonal antibody (Proteintech, 15112-1-AP, 1:500), rabbit anti-PARP polyclonal antibody (Cell Signaling, 9542S, 1:100), rabbit anti-VCP monoclonal [EPR3307(2)] antibody (Abcam, ab109240, 1:100), rabbit anti-SEC24C polyclonal antibody (Abcam, ab122633, 1:100), rabbit anti-eIF3A polyclonal antibody (Cell Signaling, 2538S, 1:50), rabbit anti-S6 ribosomal protein monoclonal (5G10) antibody (Cell Signaling, 2217S, 1:200), rabbit anti-TDP-43 polyclonal antibody (Proteintech, 10782-2-AP, 1:200), rabbit anti-IARS1 polyclonal antibody (Proteintech, 26942-1-AP, 1:50). Secondary antibodies used for IF: Donkey anti-Rabbit IgG (H + L) Secondary Antibody, Alexa Fluor 488 (Invitrogen, A21206, 1:500), Donkey anti-Mouse IgG (H + L) Secondary Antibody, Alexa Fluor 568 (Invitrogen, A10037, 1:500).

**Live cell imaging.** G3BP1-EGFP U2OS cells were plated on 4-chamber glass-bottom 35 mm dishes (Cellvis, D35C4-20-1-N) and cultured for 2 days before experiments. When cells were at 80–90% confluency, the culture medium was changed with fresh culture medium containing either DMSO or (S)-4′-nitro-blebbistatin, and cells were incubated at 37 °C with 5% $CO_2$ for 3 h. Next, cells were supplemented with 10 μg/mL of Hoechst 33342 staining solution and incubated at 37 °C with 5% $CO_2$ for 20 min. After two washes with PBS, a fresh culture medium containing stress-inducing chemicals together with either DMSO or (S)-4′-nitro-blebbistatin were added immediately before imaging. Cells were imaged using Nikon Ti2-E inverted microscope equipped with a CSU-W1 spinning disk confocal scanner (Yokogawa) and live cell maintaining system (oko lab), with NIS-Elements AR (Nikon, 5.20.00) software. Cells were maintained at 37 °C with 5% $CO_2$, and one image for each condition was captured every 2 min.

For live cell imaging of MYL9, G3BP1-EGFP U2OS cells were plated on a 6-well plate. The next day, 500 ng of each plasmid was transfected, and on the third day, cells were plated on 4-chamber glass-

bottom 35 mm dishes. The previously described live cell imaging protocol was then initiated. Only the cells exhibiting low fluorescence intensity of mCherry were analyzed for SG counting.

**Immunoblotting.** Protein concentration was determined using the BCA Protein Quantification Kit (Yeasen, 20201ES76). Samples were separated using 10% SDS-PAGE gels and transferred to a nitrocellulose membrane (Bio-Rad, 1620177) at 200 mA for 2 h. Membranes were blocked with 5% non-fat milk in TBST. Membranes were incubated with primary antibodies in Primary Antibody Dilution Buffer (Byotime, P0023A) at 4 °C overnight, washed 3 times with TBST, and incubated with secondary antibodies dissolved in Secondary Antibody Dilution Buffer (Byotime, P0023D) at room temperature for 2 h. Protein bands were visualized with a LumiQ HRP Substrate solution kit (Share-Bio, SB-WB012) using Amersham Imager 680 with IQ800 Control software (GE Healthcare, 1.2.0). Primary antibodies used for immunoblotting: rabbit anti-G3BP1 polyclonal antibody (Proteintech, 13057-2-AP, 1:5000), rabbit anti-eIF3A polyclonal antibody (Cell Signaling, 2538S, 1:1000), rabbit anti-DDX3 polyclonal antibody (Invitrogen, A300-474A, 1:2000), mouse anti LC3A monoclonal (166AT1234) antibody (Abcepta, AM1800A, 1:1000), mouse anti-LAMP1 monoclonal (H4A3) antibody (Santa Cruz, sc-20011, 1:500), rabbit anti-IARS polyclonal antibody (ABclonal, A10190, 1:2000), rabbit anti-RARS polyclonal antibody (ABclonal, A6307, 1:2000), rabbit anti-EPRS polyclonal antibody (ABclonal, A15245, 1:2000), rabbit anti-KARS monoclonal (ARC1765) antibody (ABclonal, A8648, 1:2000), rabbit anti-QARS polyclonal antibody (ABclonal, A6960, 1:2000), rabbit anti-β-Actin polyclonal antibody (Cell Signaling, 4967S, 1:1000), mouse anti-alpha-tubulin monoclonal (1E4C11) antibody (Proteintech, 66031-1-Ig, 1:20,000). Secondary antibodies used for immunoblotting: Peroxidase-AffiniPure Goat Anti-Rabbit IgG (H + L) (Jackson, 111-035-144, 1:10,000), Peroxidase-AffiniPure Goat Anti-Mouse IgG (H + L) (Jackson, 115-035-146, 1:10,000).

## Quantification and statistical analysis
Statistical details, including n and statistical tests used, can be found in the figure legends.

## Proteomic data analysis
The raw data were processed using Proteome Discoverer software (Thermo Scientific, 2.2); the UniProt human protein database (release 2022_04, 20613 sequences) was used for database search. Trypsin/P was set as the enzyme, and two missed cleavage sites of trypsin were allowed. Mass error was set to 5 ppm for precursor ions and 0.02 Da for fragment ions. Carbamidomethylation (C, + 57.02 Da) and TMT6plex (N, K, + 229.163 Da) were used as fixed modifications, and oxidation (M), deamidation (NQ), acetylation (Protein N-term) were set as variable modifications. False discovery rate (FDR) thresholds for protein and peptide were specified at 1%. The minimum peptide length was set at 7. All other parameters were set to default values for the software.

The search results were not normalized and were subjected to the ComBat method[102] from the sva R package (3.44.0; R version: 4.2.1). Initially, each column of replicates was normalized to an equal amount, which corresponded to the mean value of all columns, Combat was applied with the HS0 and HS10 time points designated as the early group, HS20 to HS60 as the middle group, and HS120 to Re10 as the late group in each replicate, the resulting combat-processed data was further normalized using the TMM normalization method[103]. The G3BP1 and CAPRIN1 datasets were processed with ComBat, respectively.

The results were then filtered based on: High protein FDR confidence (maximum of 1%); at least 1 unique peptide per replicate. For the identification of SG constituents, mitochondrial proteins, keratins, and proteins without annotated gene names were first excluded. The fold change was calculated for each protein using two approaches: (1) the maximum value observed in the HS10 to Re10 time points compared to the HS0 control (max/Ctrl), and (2) the HS0 control compared to the minimum value observed in the HS10 to Re10 time points (Ctrl/min). Proteins showed >1.2-fold change of max/Ctrl or Ctrl/min in both G3BP1 and CAPRIN1 pull-down datasets were termed as dynamic proteins. The protein intensities in the CAPRIN1 pull-down dataset were used in subsequential analysis.

The enrichment score of SG-ls was determined through a scaled fold change calculation, comparing the SG-ls values to the maximum value observed within the HS10 to Re10 time points (SG-ls/max). The fold change values were then scaled using the variance scaling method, which resulted in a z-score for each protein. Since the fold change values were centered around 0, a positive score for the scaled fold change indicates a tendency to enrich in SG-ls, while a negative score indicates a tendency to enrich in heat shock SGs. It should be noted that the SG-ls data is only in the G3BP1 pull-down dataset.

## Cluster analysis
Cluster analysis for the dynamic proteins was performed using ClustVis[49], using the Maximum distance and average linkage. The rows were scaled using the unit variance scaling prior to clustering, resulting in a z-score for each protein.

## Bioinformatics
The previously reported SG constituents list was downloaded from the RNA Granule Database 2.0[104], including data from physical studies that utilized prey-based analysis of SGs (Vu et al.[105]; Youn et al.[40]; Jain et al.[39]; Markmiller et al.[43]; Marmor-Kollet et al.[40]; Padron et al.[42]) and colocalization data with SGs. Additionally, a dataset of lost, increased, and decreased proteins in 2x Ash1-G3BP1 from a proximity-labeling study[5] was also incorporated in the reported SG proteins list.

Protein sequences were retrieved from the UniProt database. Sequence-based IDR prediction was performed using IUPred and ANCHOR[51,52]. An IDR coverage score was determined for each protein (the length of regions on a protein with >0.5 IDR score/the total length of a protein). Sequence-based prediction for PrLDs was performed using PLAAC[53].

For the construction of a protein-protein interaction network, all nodes in the network were from our list of dynamic and invariable proteins. Edge information was obtained by combining physical protein-protein interactions (confidence score >0.7) from STRING 12.0[59], BioPlex 3.0[60]. The resulting network has the largest connected component of 410 nodes and 2263 edges, which was adopted as the SG network for further visualization and analysis using Cytoscape (3.9.1)[61]. Notably, edges in the network are based on information completely independent from our study. Topological features of the network were calculated using either NetworkAnalyzer (4.4.8)[62] or NAP (2.0)[65].

GO enrichment analysis was performed using Metascape[67]. Terms with a p-value <0.05, a minimum count of 3, and an enrichment factor >1.5 are collected and grouped into clusters based on their membership similarities.

## Image quantifications
Immunofluorescence and live cell images were visualized and analyzed using Fiji ImageJ (1.53t). SGs. ROIs for individual cells were manually traced and ROIs for SGs were traced using the Analyze Particles feature in either the G3BP1 or the CAPRIN1 channel. SG number and average size in each cell were next determined using the Analyze Particles feature. For the quantification of protein-enrichment level in SGs: The G3BP1 channel was used to generate ROIs for SGs and ROIs for individual cells were manually traced. The SG-enrichment level of a target protein in a cell is calculated as its average fluorescence intensity from all SGs in a cell over its average fluorescence intensity from the entire cell. Particle analysis was performed using the Particle Tracker 2D/3D feature of the Mosaic Plugin for ImageJ/Fiji. For the quantification of SG fusion events, equal-

sized ROIs containing ~10 granules were randomly selected from blebb- and vehicle-treated live cell images. The resulting 50 blebb-treated and 50 vehicle-treated ROIs were exported as video clips and randomly shuffled before handling to two researchers to inspect whether SG fusion happened in each video clip. Results from both researchers are largely in agreement and the average result was reported.

## Statistical analysis

Statistical details, including N and statistical tests used, can be found in the figure legends. Statistical analysis was performed using GraphPad Prism (9.0) or Excel. $P \geq 0.05$ was considered as not significant (ns), $*P < 0.05$, $**P < 0.01$, $***P < 0.001$, and $****P < 0.0001$.

## Reporting summary

Further information on research design is available in the Nature Portfolio Reporting Summary linked to this article.

## Data availability

The RNA granule database is available at https://rnagranuledb. lunenfeld.ca/. The mass spectrometry data has been deposited in the ProteomeXchange member repository, under the accession code PXD044967. The source data generated in this study are provided in the Source Data file. Further information and requests for resources and reagents should be directed to and will be fulfilled by the lead contact, S.H. (hushy@shanghaitech.edu.cn). Requests of cell lines described in this study will be available upon request from the lead contact, under a standard MTA.

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

## Acknowledgements

We acknowledge the Multi-Omics Core Facility (MOCF) in the School of Life Science and Technology at ShanghaiTech University for LC-MS/MS analysis and initial data processing. We thank the Molecular and Cell Biology Core Facility (MCBCF) and the Molecular Imaging Core Facility (MICF) in the School of Life Science and Technology at ShanghaiTech University for technical support. We thank Professor Yang Yang, Professor Margaret S. Ho at ShanghaiTech University, and members of the Bai lab for their critical comments. We thank Professor Zhi Zhou at ShanghaiTech University for providing antibodies, Professor Tong Wang at ShanghaiTech University for sharing MYL9 plasmids, and Professor Wenqing Shui at ShanghaiTech University for the critical reading and constructive comments. This work has been supported by the National Science Foundation of China Grant 31670756 (to Y.B.) and Grant 32070697 (to Y.L.), the Ministry of Science and Technology of the People's Republic of China Grant 2016YFA0500901 (to Y.B.), and the start-up grant from ShanghaiTech University (to Y.B.).

## Author contributions

S.H., Y.B. and Y.L. conceived and designed the study; C.Y. generated stable cell lines; S.H., Y.Z., and Q.Y. performed the experiments; S.H. and Y.B. analyzed the data; S.H., Y.B., and Y.L. wrote and revised the manuscript, with contributions from C.Y.

## Competing interests

The authors declare no competing interests.
