## [Peer Review File · Nature Communications]

Reviewers' comments:

Reviewer #1 (Remarks to the Author):

In this manuscript, Hu et al. identify the dynamic and invariable protein components of stress granules (SGs) formed by heat shock treatment. Using U2OS cells, the authors induce SGs by incubating their cultures at 43 °C for 10, 20, 30, 60, 120, and 180 minutes, and for 5 and 10 minutes after removal from the heat shock stress. SGs were purified using the biochemical fractionation method previously developed by Roy Parker's group, and affinity purification with the core SG component G3BP1 was further employed to select for SG components. Bioinformatic analysis identified three distinct groups of SG proteins: invariable components, which were always associated with SGs; early dynamic proteins, which only associated with SGs in the first ~10-40 minutes; and late dynamic proteins, which were enriched within SGs after ~1 h. The early dynamic proteins were more enriched in RNA binding proteins (RBPs) containing intrinsically disordered regions (IDRs) and prion-like domains (PrLDs). Moreover, early dynamic proteins are more interconnected with other SG components. The authors claim that immunofluorescence staining of early and late dynamic proteins confirm the relative enrichment of proteins identified by their proteomics studies. Finally, they find that a novel SG protein, non-muscle myosin II complex (NM II), enhances SG formation and fusion events. Together, Hu et al. propose a model in which SGs are enriched with a rotating cast of interacting proteins, which shifts from RBP-containing proteins to proteins engaged in other cellular processes as the stress event lingers.

Overall, the authors' results are interesting. Although the G3BP1 interactome was previously reported (Markmiller et al. Cell 2018; Youn et al. Mol Cell 2018; Marmor-Kollet et al Mol Cell 2020), a time-resolved interactome gives the field a better understanding of how prolonged stress ensnares cellular components into persistent SGs. The topline trends reported by the authors also make sense, i.e. that IDR-containing RBPs are first recruited to SGs followed by other proteins that are more weakly incorporated into the SG interactome.

However, I have a major concern regarding the proteomic data: many the proteins identified by the interactome do not appear to be SG proteins, and they may be contaminants from the biochemical purification procedure used by the authors. In particular, the authors lack any validation that their SG fraction contains only SG components (e.g. G3BP1, TIA, TDP-43, etc.) and not proteins that are known not to engage in SGs. The high proportion of "novel" SG proteins (258 out of 506, ~50%, as noted on page 4) seems unlikely given that the SG proteome has been extensively probed before. Biochemical analyses of the "purified" SGs beyond mass spectrometry (e.g. Western blots) are essential to confirm that other cellular components, especially upregulated and/or activated heat-shock proteins, are not being co-purified with the SGs. It is also necessary to have some sort of control (e.g. mass spec of a G3BP1 pulldown without stress) to confirm that diffuse proteins are not pulled down along with the SGs.

Second, the PARP1 data is troubling. Most papers report that PARP1 is a predominantly nuclear protein, and the lone paper that the authors cite referring to cytoplasmic PARP1 used a pancreatic cancer cell line with mutations in PARP1. A more recent paper showed robust cytoplasmic translocation of WT-PARP1, but only in response to DNA damage and viral infection (Wang et al Mol Cell 2022). The first and second columns (HS20 and HS30) in Figure 5C showing the “early dynamic” localization of PARP1 to SGs appear to have cytoplasmic PARP1 that is barely above background levels. The vast majority of PARP1 in Figure 5C appears to be nuclear. Moreover, SGs were did not contain PARP1 in earlier work (Leung et al Mol Cell 2011), and a recent paper showed that neither PARP1 inhibition nor partial knockdown had any effect on SG formation (Rhine et al Mol Cell 2022). Given the rapid increase in PARP1 activity in response to stress events, it seems more likely that PARP1 was artificially pulled down in the mass spec experiments and that the PARP1-G3BP1 colocalization is an artifact of image overexposure.

Other immunofluorescence images are also not very convincing, including VCP (Figure 4D). The VCP channel appears overexposed, and it is difficult to discern whether VCP is actually enriched in SGs or not. The HS20 and HS30 SG images with SEC24C (Figure 4F) lack insets to compare G3BP1 and SEC24C. Immunofluorescence images of IARS, QARS, KARS, EPRS, and RARS are not shown in Figure 4A, and the Western blot does not include an invariable SG protein control to judge the actual enrichment of these MSC components. Overall, the validation of the proteomics data needs more rigorous experimental evidence to support the SG incorporation of these proteins, such as co-IP, FRET, etc.

Since these are the representative proteins that the authors chose to include as examples of early and late dynamic SGs, it is imperative that they clearly show an enrichment pattern showcasing the early and late dynamics they claim. The authors need to unequivocally demonstrate that they are in fact purifying SGs because I am not convinced by the proteomics data alone and the validation of certain hits via immunofluorescence. Therefore, major revisions are needed to address these concerns before this manuscript is suitable for publication in Nature Communications.

I have a few other minor concerns:

1. The authors repeatedly claim that they profiled the “full life cycle” of heat-shock SGs; however, they never probe past 10 minutes of recovery, and SGs are still present at this time point. If the authors are indeed profiling the full life cycle of an SG, they need to demonstrate when exactly heat-shock SGs are resolved and include recovery time points up until that resolution time point. The current data only shows the SG proteome in response to chronic heat-shock stress + 10 minutes, when it has been shown that SGs often take hours to resolve (Wheeler et al eLife 2016).
2. Blebbistatin is known to induce severe cytotoxicity (Mikulich et al Biochim Biophys Acta 2012), and it is unclear whether 3 h blebb treatment on its own is reducing SGs. Orthogonal methods should be used to confirm the NM II is required for SG formation, e.g. NM II knockdown/knockout.
3. Figure 1A lacks a column for HS10 even though it is included in Figure 1B.

4. The significance markers on Figure 2E are confusing, and I am not sure what is being compared.
5. There is a typo on page 10 (PAPRs)

Reviewer #2 (Remarks to the Author):

SUMMARY

Hu et al. expands upon previous efforts to characterize the stress granule (SG) core proteome throughout a prolonged heat shock stress-and-recovery time course. The authors use SG purification method established by Jain et al., Mol Cell 2016, which involves differential centrifugation of lysates followed by G3BP1 immunoprecipitation to systematically sample SG proteins. Their proteomics analysis identified 506 SG proteins, about half of which were binned to invariable group or the other to a dynamic group where the proteins are accumulated differentially across time points. The dynamic group was divided into “early dynamic” SG proteins, described by a high capacity to phase separate (high isoelectric point, IDR content, % RBPs), high degree centrality within the SG network, and RNA-centric functions, and “late dynamic” SG proteins, described by significantly lower capacity to phase separate, lower degree centrality but instead modular in their network, and more diverse functions. To determine the nature of their recruitment to SGs, additional proteomic analysis was performed on SG-like condensates (SG-Is) that can be generated by increasing the protein levels of SG nucleator, G3BP1. Comparison to the SG-Is proteome showed that early dynamic SG proteins were far more likely to be enriched in these SG-Is generated by G3BP1.

Next, they validated the localization of individual dynamic proteins to SGs throughout prolonged heat stress and recovery using immunofluorescence imaging of heat shock treated cells for varied duration of time. Proteomics analysis support that late dynamic SG proteins are recruited to SGs together as functional modules, such as eukaryotic initiation factor 3 (eIF3), the multisynthetase complex, or the COPII coat complex. Conversely, early dynamic proteins, such as RPS6 or PARP1, lose their association with SGs at some point prior to SG disassembly. Additionally, they report novel functions of subunits of the non-muscle myosin II complex (NM II), motor proteins known to work along actin cytoskeletons. Identified as early proteins, authors find that NM II proteins promote SG formation, fusion, and mobility during heat shock and osmotic stress conditions.

The authors survey and classify heat-induced SG-enriched fraction during prolonged stress and recovery using quantitative proteomics method to obtain a temporal resolution in SG proteome. Their work is a

strong attempt to investigate the dynamic changes that occur in SGs over the course of stress response and adaptation. Similar time-resolved studies have been published using APEX before (Padron et al., 2019 and Marmot-Kollet et al., 2020). However, compared to previously published work, this system offers a systematic evaluation of time-gated compositional changes that occur during prolonged cellular stress and offers evidence for stress-induced modalities. This dataset is novel and has the potential to be a useful resource to better understand distinct recruitment kinetics of specific SG proteins. However, this study lacks important control samples for the initial proteomics analysis and generally lacks transparency and assessments needed to validate the quality of presented data. Also, I strongly disagree that authors can generate SG-like condensate at steady state. This contradicts what others have reported previously. Lastly, use of a single protein to enrich for SG proteins (G3BP1-IP) is not enough to survey temporal changes at the 'proteome-wide' level. The rationale for using G3BP1 to enrich for SGs is also weak, given that G3BP1 is not required for SG formation under heat shock stress.

MAJOR

1. This study lacks important control experiments to quantitatively determine the time-resolved changes in SG protein abundance. The current manuscript also fails to show transparency in their experimental workflow and analysis of quantitative proteomics data.

a. Authors use total cell lysates as the sole control for quantitative assessment of G3BP1-associated SG proteins. This control alone is insufficient to assign identified proteins from stress timepoints as a 'SG' protein. Assignment of a 'SG' protein should include additional controls.

First, authors need to include a 'pre-heat shock' time point that has undergone the same SG-enrichment and G3BP1 immunoprecipitation steps. This is an important reference timepoint to determine early dynamic, late dynamic and invariable components associated with G3BP1. This timepoint will represent G3BP1-associated proteins from 'SG fraction' when microscopic SGs are not detectable. Dynamic 'early' proteins should be defined by those that increase in abundance relative to this HS 0 timepoint.

Please clarify if 'G3BP1-IP' is referring to standard IP or IP after 'SG fraction' enrichment via differential centrifugation. If latter, this should be referred to as 'HS 0' and used as a control.

Second, authors fail to acknowledge the influence of prolonged heat shock on protein solubility and need to incorporate changes in the 'input' material used for G3BP1-IP. Heat shock induces protein aggregation, which will precipitate during differential centrifugation (see PMID: 33022891). Protein aggregation will cause the input 'SG fraction' to vary throughout the time-course. Authors need to perform TMT quantitative analysis of the 'SG fraction' used for all timepoints to account for this variability. Authors can then use this 'input' as control and determine enrichment scores for each protein across timepoints. This is crucial to accurately determine the relative abundance of SG proteins in each time point. Without this quantitation, some of 'dynamic' SG proteins may simply be reflection of

proteins that are differentially aggregated or solubilized by heat shock proteins during the prolonged heat stress, whereas 'invariable' proteins are those that are changing.

I have concerns about the specificity and stringency to claim 'SG' proteome used in this current version. Many proteins reported in the supplementary table as SG proteins are unlikely candidates to localize to SGs (histones, membrane associated proteins like OSBPL10 and OR5AC2). This is likely due to lack of controls and inadequate data analysis and will improve by inclusion of additional controls mentioned above.

b. Unclear data analysis workflow. Authors have not provided clear information on how input peptide material were normalized, how quantified protein abundance were normalized, and how z-score were calculated. How did authors resolve the variability in input material and TMT labeling efficiency between batches? Authors did note that protein abundance was normalized to '1,000,000', but what normalization method was used? (e.g., Linear or quantile?) This needs to be included as supplementary information. How is z-score calculated? The mean abundance of protein and variation used for z-score – are these derived from multiple timepoints? If so, proteins whose abundance varies greatly between timepoints will be penalized in this scoring regime, which may be of most interest for biological context. Lastly, we were unable to read the supplementary table provided in the PDF format, as columns of numbers are presented without corresponding gene names. They should be in tabulated format along with raw data.

2. Here authors show that increasing total amount of G3BP1 by two-fold was sufficient to induce SGs without any stress. This is hard to believe and contradictory to other publications showing that no SGs are detectable in no stress condition, regardless of the concentration of overexpressed G3BP1 (Figure 1G in Sanders et al., 2020). Many others have representative microscopy images showing absence of SGs in no stress condition. Authors need to state this discrepancy and include quantification of SGs observed in experiment presented in Fig 2D.

3. This manuscript does not provide means to assess the quality of the 'SG proteome' defined by their proteomics data. For instance, they fail to show how their identified proteome compares to other datasets. There are SG proteomes defined through literature curation (Youn et al., 2019) or using the same method (Jain et al.,) and other methods (APEX- Marmor-Kollet et al., 2020, Markmiller et al., 2018 and BioID – Youn et al., 2018). Validation of a few 'early' and 'late' proteins is not sufficient to assess the quality of the data.

4. Authors need to acknowledge the limitation of G3BP1-centric purification of SG core using this method. Specifically, it is limited in terms of the coverage of the SG proteome. Authors should state that

the data shows G3BP1 associated proteins in SG-enriched fraction, rather than presenting this data as a comprehensive 'SG proteome' [e.g., Line 429]. I strongly recommend immunoprecipitating other SG proteins (those validated to localizes to SGs during heat shock) to improve the coverage and specificity of the method used.

5. 'Late' proteins include components of translational machineries [translation initiation factors (EIF4A1, EIF2S1) and eIF3 complex components], which are expected to be invariable or recruited early, given that SGs arise from translational machineries. EIF3A in Fig 4B clearly shows that it resides in SGs at the earliest timepoint. To propose that these translational factors as 'late' proteins, authors need to correct their conclusion and include validation of additional proteins in this complex.

6. [Figure 6] Genetic methods to chemical inhibition would be necessary to support the function of type II NM in SG fusion.

MINOR

1. Please rephrase/ change the interpretation of these conclusions in 'early dynamic' SG proteins.

a. [Line 71-72, 394-396] The proposed model suggests that the RNA-centric functions of "early dynamic SG proteins" bespeaks a role in RNA-related cellular processes during the initial stages of SG formation, however, they do not address the possibility that their recruitment to SGs is purely a consequence of their potent phase separation tendencies. Please do not assign function based on correlative observation without supporting data.

b. [Line 193-195] The high degree centrality of early proteins does not necessitate that they "play essential roles in holding the entire SG network together". In fact, their high centrality may be a consequence of their greater IDR content, which might facilitate more transient (albeit non-essential) interactions with other proteins. Again, cannot assign function based on network analysis without supporting data.

c. [Line 37] The disruption of SG dynamics has never been shown to directly cause neurodegenerative diseases; the pathological inclusions mentioned only mimic those in true disease contexts.

2. It is not specified if the 1.2-fold change required to categorize a protein as "dynamic" is measured between consecutive timepoints, or between any two points of the time course.

3. [Line 104-105, Fig 1C] It is not specified what sources were relied upon to determine a protein's prior annotation as a SG constituent.

SIDE NOTE

It is interesting that late proteins tend to have lower isoelectric point. Could this be related to the mechanism where brief acidification of the cytosol occurs in various stress conditions (including heat shock) induces stress response in eukaryotic cells, as shown in studies by Triandafillou et al., eLife 2020, Weitzel et al., Experimental Cell Research 1987; Bright and Ellis, Experimental Physiology, 1992; Munder et al., eLife 2016; Kroschwald et al., eLife 2015? If cytosol is acidified during late time points in the heat shock, this would neutralize proteins with lower isoelectric points, causing them to phase separate or aggregate.

Typographical Corrections

[Line 32] “Therefore” is an inappropriate conjunctive adverb here.

[Line 178] A comma should follow the word “Previously”.

[Line 204] “Constituent” should be changed to either “constituency” or “constituents”.

[Line 302 & 305] “PAPR” should be changed to “PARP” (poly-ADP-ribosyl polymerase)

[Line 306 & 307] “Nuclear” should be changed to “nucleus”.

[Line 309] “Becoming diffused” should be changed to “becoming diffuse” or simply “diffusing”.

[Line 321] “As well as the gathering of...” (addition)

[Line 322] “The microtubule network and its associated motors have been reported to be involved in...” (addition & corrections)

[Line 360-361] “As a result” is an inappropriate prepositional phrase here.

[Line 361] “Happen” should be changed to “happens”.

[Line 366] “Promote” should be changed to “promotes”.

[Line 401-402] “recruitment of these modules seems to be independent of each other, supported by the...” (correction & deletion)

[Line 426] “the NM II complex” (addition, keeping format with Line 416)

Reviewer #3 (Remarks to the Author):

Stress granules (SGs) are dynamic, phase-separated RNA-protein condensates that form in cells in response to various cellular insults. In recent years, there have been several efforts to catalog the protein constituents of stress granules through proteomic approaches. While it is known that some properties of granules can change over the course of prolonged stress and recovery, potential time dependent changes in stress granule make-up have been largely unexplored. In this manuscript, Shuyao Hu et al. adapt previously used methods to explore dynamic changes in the stress-granule profile during the cycle of heat shock and recovery. The authors show that while a substantial portion of the stress granule proteome remains constant over time, many proteins have a dynamic profile, accumulating at either early or late time points in the heat stress process. Interestingly, the authors provide data to suggest that early and late SG proteins have distinct properties, are recruited to SGs as functional modules and may serve unique roles in regulating SGs. This work is quite novel, as little has been published about how SG profiles might change over time and is therefore likely to be of significant interest to the field. Furthermore, the authors provide a specific example in the case of a myosin module regulating stress granule formation which highlights how their proteomics data set can provide new insight into stress granule regulation for future studies. The conclusions are generally well supported, with cell imaging and immunoblot data helping to validate the proteomics. Notably, there is potentially a significant issue in the way the authors have defined stress granule constituents based on the proteomics data that would need to be addressed prior to publication, along with several more minor points listed below. If the authors can sufficiently address these points in a revised manuscript, this study would likely have a significant impact on the field of SG biology.

1. One major point that needs to be clarified is how the authors define SG constituents based on their proteomics data, as all the subsequent results are dependent on this analysis. In the methods section on starting on line 729, the authors state that proteins were considered SG constituents if they met one of the following conditions in “both replicate series”: (1) detected in only SG samples, (2) enriched >2 fold in SG versus total protein samples. Many of the proteins identified as “dynamic SG proteins” in Table 1 show enrichment in only in one of 2 replicates (e.g. AARS, ZNF638, WDR33) or enrichment in neither replicate (e.g. ANXA11, TUBB4B, NHP2). It is somewhat confusing that these examples are termed dynamic SG proteins even though they don’t fit the enrichment versus total protein criteria of “SG constituents” as defined for the invariable SG proteins. As written, it seems that a protein would be counted as a dynamic SG protein even if it were increased at a given stress time points in both the total protein fraction and SG fraction, with no enrichment in the SG fraction. However, such a protein may be better described as being dynamically globally upregulated during stress, independent of any recruitment to SGs (notably this appears to be the case for AIMP1 in replicate 2) and thus should not be considered a dynamic SG constituent. Furthermore, it is unclear if the authors have used sample with G3BP-IP from unstressed cells in determining a proteins status as an SG constituent. Previous studies using MS approaches to identify SG constituents (e.g. Ref38, Jain et. al), enrichment in stressed vs unstressed G3BP IP samples was used as a criteria rather than enrichment versus total protein level. The authors need to clarify their criteria and either adjust their list of dynamic SG proteins or make a case as to why they have included these proteins in a list of SG constituents rather than simply proteins for which the total level is dynamically regulated during stress.

Minor Issues

2. The conclusions from the data concerning recruitment to stress-independent G3BP1 granules (SG-Ls) are potentially interesting, but the data presentation and discussion are somewhat difficult to follow. The authors define the enrichment score as a the “relationship” of z scores for unstressed versus stress-induced granules. The calculation of this enrichment score is unclear to this reviewer, such that I’m not sure what it means for a protein to have an SG-Ls enrichment score of zero. The authors state that positive scores indicate HS-independent recruitment which I would interpret to mean the protein has similar levels in SGs and SG-Ls, but without seeing how the value is calculated, one might also assume that a score of zero might indicate no difference in levels between SGs and SG-Ls. Perhaps the author’s could present the analysis as a of enrichment between SGs and SG-Ls that is more immediately interpretable? Also, the data in Fig. 2E might be easier to interpret as a frequency distribution histogram rather than a cumulative distribution.

3. The author’s show a comparison between SG-Ls and G3BP1-IP samples, but do not seem to mention any comparison of the stress-independent G3BP1-IP data to the identified groups of SG constituents. The authors should include some analysis to show how protein levels in the unstressed G3BP1 Ip compare to levels detected in the SGs and use this as a factor in determining whether or not a protein should be considered a bona fide SG constituent, either of the dynamic or invariable variety.

4. The authors might consider including an addition simple table with lists of proteins in each determined category (i.e. invariable, early and late SG proteins). This would allow readers to more easily browse the make-up of each protein type.

5. The authors identify as late dynamic SG components. Figure 4B shows that eIF3A does appear to have increased colocalization with G3BP1 at later time points. Interestingly however, the eIF3A signal is clearly observed in puncta at the earliest time point, some of which appear to be G3BP1 negative. This observation highlights one limitation of defining SGs by a single marker (e.g. G3BP1) and might suggest that eIF3A is actually recruited to SGs prior G3BP1, or instead that eif3A is recruited to SG early but is also found in non-SG puncta early in stress (e.g. p-bodies). Also the authors state that “eIF3 subunits are classified as late... Line244”, but the MS data shows that 5 eIF3 subunits are also included in the invariable category. The authors should discuss these issues more carefully.

6. It might be helpful to show fixed-cell images of cells prior to any HS at the resting state so readers can better appreciate the HS-induced puncta formation.

**Reviewer #1:**

We would like to thank this reviewer for his/her careful reading of the revision and the
critical and constructive comments. We extend our gratitude to the reviewer for
acknowledging the significance of our time-resolved interactome study, which
contributes to a deeper understanding within the field of how prolonged stress captures
cellular components within persistent SGs. The primary concern raised by this reviewer
pertains to the potential contamination of SG components by nonspecific proteins
under our experimental conditions, as well as the need for essential controls to
eliminate background interference. In response to this valid concern, we have taken
several measures in this revision. Firstly, we introduced a control group subjected to
no stress conditions (referred to as HS0) to serve as a baseline comparison.
Additionally, we conducted a thorough bioinformatic data analysis to ensure the
reliability of our results. Furthermore, we have included pivotal immunoblotting results
that serve to validate the purity of SG components within our experimental setup.
Below is our point-by-point response to each comment. Changes to the text are
highlighted in red.

In this manuscript, Hu et al. identify the dynamic and invariable protein components of
stress granules (SGs) formed by heat shock treatment. Using U2OS cells, the authors
induce SGs by incubating their cultures at 43 °C for 10, 20, 30, 60, 120, and 180
minutes, and for 5 and 10 minutes after removal from the heat shock stress. SGs were
purified using the biochemical fractionation method previously developed by Roy
Parker's group, and affinity purification with the core SG component G3BP1 was
further employed to select for SG components. Bioinformatic analysis identified three
distinct groups of SG proteins: invariable components, which were always associated
with SGs; early dynamic proteins, which only associated with SGs in the first ~10-40
minutes; and late dynamic proteins, which were enriched within SGs after ~1 h. The
early dynamic proteins were more enriched in RNA binding proteins (RBPs) containing
intrinsically disordered regions (IDRs) and prion-like domains (PrLDs). Moreover, early
dynamic proteins are more interconnected with other SG components. The authors
claim that immunofluorescence staining of early and late dynamic proteins confirm the
relative enrichment of proteins identified by their proteomics studies. Finally, they find
that a novel SG protein, non-muscle myosin II complex (NM II), enhances SG
formation and fusion events. Together, Hu et al. propose a model in which SGs are
enriched with a rotating cast of interacting proteins, which shifts from RBP-containing
proteins to proteins engaged in other cellular processes as the stress event lingers.

Overall, the authors' results are interesting. Although the G3BP1 interactome was
previously reported (Markmiller et al. Cell 2018; Youn et al. Mol Cell 2018; Marmor-
Kollet et al Mol Cell 2020), a time-resolved interactome gives the field a better
understanding of how prolonged stress ensnares cellular components into persistent
SGs. The topline trends reported by the authors also make sense, i.e. that IDR-
containing RBPs are first recruited to SGs followed by other proteins that are more

weakly incorporated into the SG interactome.

**Major concerns:**

1. Many the proteins identified by the interactome do not appear to be SG proteins,
and they may be contaminants from the biochemical purification procedure used by
the authors. In particular, the authors lack any validation that their SG fraction
contains

97 only SG components (e.g. G3BP1, TIA, TDP-43, etc.) and not proteins that are known
not to engage in SGs. The high proportion of “novel” SG proteins (258 out of 506,
~50%, as noted on page 4) seems unlikely given that the SG proteome has been
extensively probed before. Biochemical analyses of the “purified” SGs beyond mass
spectrometry (e.g. Western blots) are essential to confirm that other cellular
components, especially upregulated and/or activated heat-shock proteins, are not
being co-purified with the SGs.

Response:

we thank the reviewer for raising this critical issue. In our updated datasets, which
include time-series data in comparison to the non-heat shock (HS0) control, the
proportion of previously reported SG proteins is about 75% (290/385), confirming the
consistency of our data with previously studies (depicted below, and please also refer

to the response to the following question for more details).

We have further performed immunoblotting experiments to evaluate SG fractions
under both non-stress conditions and 180-min heat shock, utilizing CAPRIN1 as an
SG enrichment bait (Figure S1B and depicted below). The results unambiguously
demonstrate a significant enrichment of three well-defined SG markers—namely
G3BP1, eIF3A, and DDX3X—in the samples subjected to heat shock, as compared to
those maintained under non-stress conditions. Conversely, neither the induction of

heat shock nor the absence of stress led to an augmentation of autophagosomes and
lysosomes within the SG fractions. These data suggest the authenticity of SG
components under our experimental conditions.

Figure S1B. Immunoblots of the total input and the SG fraction samples collected
 under normal growth conditions or heat shock.

When it comes to heat shock proteins (HSPs), it's worth noting that a multitude of them,
 such as HSP90AA1, HSPA9, HSPB1, and HSPD1, have already been identified as
 components of SGs. These identifications have been established using various
 methods, including G3BP1 pull-down (Jain et al. 2016), proximity-dependent
 biotinylation (Markmiller et al. 2018; Marmot-Kollet et al. 2021; Padron A et al., 2019),
 and colocalization with SGs, as documented in the RNA Granule Database 2.0 (Millar
 SR. et al., 2023). In light of these existing findings, our hypothesis emerges that HSPs
 might not be suitable candidates for serving as non-SG protein markers. The exclusion
 of autophagosomes and lysosomes in SG fraction serves as a strong illustration of the
 effectiveness of our SG marker pull-down approach, showcasing minimal
 contamination.

2. It is also necessary to have some sort of control (e.g. mass spec of a G3BP1
 pulldown without stress) to confirm that diffuse proteins are not pulled down along with
 the SGs.

Response:

We thank the reviewer for raising this critical issue. We agree with the reviewer
 regarding the significance of illustrating authentic components within the SGs.
 To address this concern, we prepared a new set of samples-HS0, HS10, and HS60-by
 applying the same differential centrifugation steps prior to G3BP1 pull-down as the
 original experiment. These data were integrated into the original G3BP1 pull-down
 dataset using the ComBat processing of sva R package (Johnson, W.E. et al.,
 Biostatistics, 2007). Subsequently, we have acquired a dataset comprising a control
 group under non-stress condition and a series of time points with varying heat-shock
 treatment durations.

Importantly, in order to enhance the specificity of our SG proteomic data, we have
 obtained an additional dataset encompassing a time series spanning from HS0 to
 Re10, utilizing CAPRIN1 for the pull-down procedure.

In particular, proteins that exhibited >1.2-fold changes over the HS0 control in at least
one time point, in both the G3BP1 and CAPRIN1 pull-down datasets, were classified
as dynamic proteins. Furthermore, within the subset of proteins with fold changes
below 1.2, 85 out of 142 were previously reported as SG constituents. We termed these
proteins as invariable proteins. Remarkably, within the 300 dynamic and 85 invariable
proteins, 290 (75%) were previously reported as SG proteins (as shown in Figure 1C
and depicted below).

Figure 1C. Bar graph shows different groups of proteins.

Notably, the analysis results of our newly combined dataset, derived from two distinct
SG marker pull-down experiments, strongly align with the conclusions drawn from our
initial dataset. The SG dynamic proteins showed a remarkably dynamic temporal
behavior throughout the heat shock process.

We have extensively revised the specific numbers and descriptions in our manuscript
to reflect the changes in input data. Changes in the text are highlighted in red.

3. Second, the PARP1 data is troubling. Most papers report that PARP1 is a
predominantly nuclear protein, and the lone paper that the authors cite referring to
cytoplasmic PARP1 used a pancreatic cancer cell line with mutations in PARP1. A
more recent paper showed robust cytoplasmic translocation of WT-PARP1, but only in
response to DNA damage and viral infection (Wang et al Mol Cell 2022). The first and
second columns (HS20 and HS30) in Figure 5C showing the “early dynamic”
localization of PARP1 to SGs appear to have cytoplasmic PARP1 that is barely above
background levels. The vast majority of PARP1 in Figure 5C appears to be nuclear.
Moreover, SGs were did not contain PARP1 in earlier work (Leung et al Mol Cell 2011),
and a recent paper showed that neither PARP1 inhibition nor partial knockdown had
any effect on SG formation (Rhine et al Mol Cell 2022). Given the rapid increase in
PARP1 activity in response to stress events, it seems more likely that PARP1 was
artificially pulled down in the mass spec experiments and that the PARP1-G3BP1
colocalization is an artifact of image overexposure.

Other immunofluorescence images are also not very convincing, including VCP (Figure
4D). The VCP channel appears overexposed, and it is difficult to discern whether VCP
is actually enriched in SGs or not.

Response:

We thank the reviewer for raising this issue. We acknowledge the reviewer's
observation regarding the predominant localization of PARP1 in the nucleus, as
reported in numerous studies. Nevertheless, we would like to highlight that two specific
studies have provided evidence of PARP1's presence within granules induced by
C9orf72 dipeptide repeats (Lee et al., 2016) or saturated fatty acids (Zhang et al.,
2021). Additionally, Yang et al. reported that PARP1 was found in G3BP1-WT SGs and
exhibited an increase in G3BP1-2x Ash1 IDR SGs (Yang et al., 2020). These findings
suggest that the limited cytoplasmic localization of PARP1 might be crucial for its
recruitment into SGs during stress conditions.

We completely agree with the concerns raised by this reviewer regarding the
overexposure of PARP1 in our images. Due to its predominant nuclear localization, we
found it necessary to increase the laser intensity in order to capture the PARP1 signal
in the cytoplasm. Nevertheless, it's important to highlight that the nuclear signals were
not subjected to overexposure. The enhanced contrast of PARP1 was solely intended
to improve the visibility of signals within the SGs for a more distinct presentation.

As demonstrated in the bottom lane of Figure 5C and the accompanying raw images
below, the green signals representing PARP1 in the cytoplasm were not saturated;
they remained well within the measurable range. Similarly, the signal profile of PARP1
in the nucleus also did not reach the saturation point.

Likewise, the images of VCP were not subjected to overexposure, and an example of
the signal profile of VCP in the nucleus and SGs is displayed below:

This confirms that the imaging conditions were carefully optimized to avoid
overexposure. In order to distinctly illustrate the presence of PARP1 and VCP
within SGs while omitting them in the nuclear region, we have retained our figures as

7

presented in the original manuscript.

Furthermore, within our CAPRIN1 pull-down data, we consistently observed the
enrichment of PARP1, along with an apparent early group trend. These findings
provide compelling evidence that PARP1 is not a nonspecifically identified SG
component.

To further address this reviewer's concern, we have included additional
immunofluorescence data to provide additional evidence through the analysis of
another early dynamic protein, TDP-43. Its fluorescence enrichment in SGs perfectly
aligns with the early pattern observed in the pull-down data. This finding further
strengthens our conclusion regarding the specific and timely recruitment of SG
components during the stress response. These new results are now presented in
Figure 5E-F.

4. The HS20 and HS30 SG images with SEC24C (Figure 4F) lack insets to compare
G3BP1 and SEC24C.

Response:

We thank the reviewer for the suggestion. We have added the insets of HS20 and
243 HS30 images in Figure 4F.

5. Immunofluorescence images of IARS, QARS, KARS, EPRS, and RARS are not
shown in Figure 4A, and the Western blot does not include an invariable SG protein
control to judge the actual enrichment of these MSC components.

Response:

We appreciate the reviewer for raising this critical issue. We have added CAPRIN1 as
invariant SG component controls in Figure 4A. Notably, CAPRIN1 consistently
appeared within the SG fraction throughout the heat shock time series, thereby serving
as a reliable reference point.

Furthermore, we selected IARS1 for colocalization analysis with SGs using
immunofluorescence. Notably, IARS1 shows an increased enrichment within SGs over

the course of the heat treatment, as depicted below and appended to Figure S4C-D:

Minor concerns:

1. The authors repeatedly claim that they profiled the “full life cycle” of heat-shock SGs;
however, they never probe past 10 minutes of recovery, and SGs are still present at
this time point. If the authors are indeed profiling the full life cycle of an SG, they need
to demonstrate when exactly heat-shock SGs are resolved and include recovery time
points up until that resolution time point. The current data only shows the SG proteome
in response to chronic heat-shock stress + 10 minutes, when it has been shown that
SGs often take hours to resolve (Wheeler et al eLife 2016).

Response:

We appreciate the reviewer for raising this point. We agree with the reviewer that SGs
persist even after the removal of stress, signifying a sustained existence beyond the
duration covered by our time points. In order to accurately reflect this aspect, we have
revised our terminology from "full life cycle" to "acute to prolonged life cycle of heat-
shock SGs." This revised description accurately captures the temporal dynamics of
SGs during the 10-min to 180-min heat shock period and the early stage of the
recovery process.

2. Blebbistatin is known to induce severe cytotoxicity (Mikulich et al Biochim Biophys
Acta 2012), and it is unclear whether 3 h blebb treatment on its own is reducing SGs.
Orthogonal methods should be used to confirm the NM II is required for SG formation,
e.g. NM II knockdown/knockout.

Response:

We appreciate this reviewer's insightful suggestion aimed at strengthening the
conclusion of our study. We sought to examine the effects of NM II knockdown by
utilizing siRNAs targeting MYH9. However, we encountered challenges as the cells
exhibited morphological damage or abnormal aggregation of the GFP signal, as shown
in the results below. These observations are consistent with other studies that suggest
the substantial inhibition of cell proliferation and migration upon knockdown of the
heavy chains of NM II (MYH9 or MYH10) (Meng Chen et al., Cell Death Discov, 2021;
Longyang Liu et al., Adv Sci (Weinh), 2023). This outcome could potentially be
attributed to the essential role that NM II plays in material transport processes. As a
result, the exploration of NM II via knockdown or knockout approaches posed
challenges, given their substantial impact on cell survival and cellular morphology.

We then investigated the role of NM II in SG assembly by activating the NM II complex. To achieve this, we transfected cells with a phosphomimetic MYL9 variant containing a T18D/S19D mutation, which mimics the phosphorylated state of MYL9 and activates myosin functions (Jordan R Beach et al., BMC Cell Biol, 2011). Live cell imaging revealed that MYL9-T18D/S19D significantly accelerated SG formation when compared to the wild-type MYL9. These results strongly support the crucial role of NM II in SG formation, which aligns with the inhibitory effects observed with Blebbistatin. We have added this result into Figure 6G and Supplementary Movie S2. Additionally, the original Figure 6G was moved to Figure S5E.

Figure 6G. Quantification of SG number per cell at different osmotic stress time.

3. Figure 1A lacks a column for HS10 even though it is included in Figure 1B.

Response:

We sincerely appreciate the valuable suggestion provided by this reviewer. In our initial version, given the limited number of microscopically distinguishable SGs observed at the 10-minute HS time point, we opted to designate the 20-minute HS time point as the earliest feasible time point for our immunofluorescence imaging studies. This decision was made to ensure optimal visualization and analysis of SGs.

Nevertheless, it is important to note that we have indeed conducted immunofluorescence imaging of G3BP1 at the HS10 time point. In the current revision, we have included the image result of HS10 in Figure S1A. By incorporating the additional images, Figure S1A provides further validation of the time course of SG

formation during the early stages of heat shock.

Figure S1A. Confocal micrographs of U2OS cells fixed at the indicated time points during HS and recovery.

4. The significance markers on Figure 2E are confusing, and I am not sure what is
being compared.

**Response:**

We thank the reviewer for pointing this out. We have replaced the original plot in Figure
2E with a scatter plot accompanied by a frequency distribution histogram. We hope the
new data presentation is not confusing.

5. There is a typo on page 10 (PAPRs)

Response:

Thank you for pointing this out. The typo has been corrected.

We thank this reviewer for the insightful comments, which have helped us substantially

improve our manuscript.

**Reviewer #2**

We would like to thank this reviewer for his/her careful reading of the manuscript and
the constructive remarks. We appreciate this reviewer for recognizing the significance
of our system, which offers a systematic assessment of time-gated compositional
changes occurring during prolonged cellular stress, while also providing insights into
stress-induced modalities. A key concern raised by the reviewer is the adequacy of
controls in the initial proteomics analysis and the overall transparency and validation
of the presented data. Additionally, the reviewer points out that the utilization of a single
protein (G3BP1-IP) for enriching SG proteins might not fully capture temporal changes
on a 'proteome-wide' scale. In response to these valuable comments, we have made
several improvements. Firstly, we have included the HS0 timepoint as a control for the
G3BP1 pull-down procedure. Moreover, we have provided comprehensive details of
all bioinformatic analyses conducted on the data. To address the concern of a single
protein-based approach, we have carried out an additional SG protein CAPRIN1 pull-
down and have compared the dataset with that obtained from G3BP1 pull-down. Below
is our point-by-point response to each comment. Changes to the text are highlighted
in red.

The authors survey and classify heat-induced SG-enriched fraction during prolonged
stress and recovery using quantitative proteomics method to obtain a temporal
resolution in SG proteome. Their work is a strong attempt to investigate the dynamic
changes that occur in SGs over the course of stress response and adaptation. Similar
time-resolved studies have been published using APEX before (Padron et al., 2019
and Marmot-Kollet et al., 2020). However, compared to previously published work, this
system offers a systematic evaluation of time-gated compositional changes that occur
during prolonged cellular stress and offers evidence for stress-induced modalities. This
dataset is novel and has the potential to be a useful resource to better understand
distinct recruitment kinetics of specific SG proteins. However, this study lacks important
control samples for the initial proteomics analysis and generally lacks transparency
and assessments needed to validate the quality of presented data. Also, I strongly
disagree that authors can generate SG-like condensate at steady state. This
contradicts what others have reported previously. Lastly, use of a single protein to
enrich for SG proteins (G3BP1-IP) is not enough to survey temporal changes at the
'proteome-wide' level. The rationale for using G3BP1 to enrich for SGs is also weak,
given that G3BP1 is not required for SG formation under heat shock stress.

**Major concerns:**

1. This study lacks important control experiments to quantitatively determine the time-
resolved changes in SG protein abundance. The current manuscript also fails to show
transparency in their experimental workflow and analysis of quantitative proteomics
data.

a. Authors use total cell lysates as the sole control for quantitative assessment of

G3BP1-associated SG proteins. This control alone is insufficient to assign identified
proteins from stress timepoints as a 'SG' protein. Assignment of a 'SG' protein should
include additional controls.

First, authors need to include a 'pre-heat shock' time point that has undergone the
same SG-enrichment and G3BP1 immunoprecipitation steps. This is an important
reference timepoint to determine early dynamic, late dynamic and invariable
components associated with G3BP1. This timepoint will represent G3BP1-associated
proteins from 'SG fraction' when microscopic SGs are not detectable. Dynamic 'early'
proteins should be defined by those that increase in abundance relative to this HS 0
timepoint.

Please clarify if 'G3BP1-IP' is referring to standard IP or IP after 'SG fraction'
enrichment via differential centrifugation. If latter, this should be referred to as 'HS 0'
and used as a control.

Response 1-a.1:

We thank the reviewer for raising these important issues. Regarding the control issue,
we agree with the reviewer that the inclusion of a negative control at the HS0 timepoint
is essential for accurately assessing the quantitative changes in SG protein abundance
over time.

In our study, the term 'G3BP1-IP' refers to standard IP of G3BP1 in untreated cells.
Therefore, it is not suitable to be used as the 'HS0' control. In order to address the
raised concern regarding controls, we took the following approach: we prepared a new
set of samples-HS0, HS10, and HS60-by applying the same differential centrifugation
steps prior to G3BP1 pull-down as the original experiment. These data were integrated
into the original G3BP1 pull-down dataset using the ComBat processing of sva R
package (Johnson, W.E. et al., Biostatistics, 2007). Subsequently, we have acquired a
dataset comprising a control group under non-stress condition and a series of time
points with varying heat-shock treatment durations.

Furthermore, to enhance the comprehensiveness of our SG proteomic data for a more
extensive exploration of temporal changes approaching a 'proteome-wide' scale, we
obtained an additional dataset encompassing a time series spanning from HS0 to
Re10, utilizing CAPRIN1 for the pull-down procedure. In order to refine the specificity
of our SG proteomic data, we employed the intersection of the G3BP1 pull-down
dataset and the CAPRIN1 pull-down dataset, rather than simply collecting the two
datasets.

In this context, proteins that exhibited >1.2-fold changes over the HS0 control in at
least one time point, in both G3BP1 and CAPRIN1 pull-down datasets, were
designated as dynamic proteins. Conversely, proteins that displayed fold changes
below 1.2, 85 out of 142 were previously reported as SG constituents, we termed these
proteins as invariable proteins (as shown in Figure 1C and depicted below).

Figure 1C. Bar graph shows different groups of proteins.

Notably, the analysis results of our newly combined dataset, derived from two distinct
SG marker pull-down experiments, strongly align with the conclusions drawn from our
initial dataset. The SG dynamic proteins showed a remarkably dynamic temporal
behavior throughout the heat shock process.

We have extensively revised the specific numbers and descriptions in our manuscript
to reflect the changes in input data. Changes in the text are highlighted in red.

Second, authors fail to acknowledge the influence of prolonged heat shock on protein
solubility and need to incorporate changes in the 'input' material used for G3BP1-IP.
Heat shock induces protein aggregation, which will precipitate during differential
centrifugation (see PMID: 33022891). Protein aggregation will cause the input 'SG
fraction' to vary throughout the time-course. Authors need to perform TMT quantitative
analysis of the 'SG fraction' used for all timepoints to account for this variability. Authors
can then use this 'input' as control and determine enrichment scores for each protein
across timepoints. This is crucial to accurately determine the relative abundance of SG
proteins in each time point. Without this quantitation, some of 'dynamic' SG proteins
may simply be reflection of proteins that are differentially aggregated or solubilized by
heat shock proteins during the prolonged heat stress, whereas 'invariable' proteins are
proteins are those that are changing.

Response 1-a.2:

We thank the reviewer for pointing out this issue. Upon further investigation, we
compared our dynamic proteins with the aggregators identified in the study conducted
by Tomi A Määttä et al. (2020). Interestingly, we found that only 15 of our dynamic
proteins were classified as aggregators, with 10 belonging to the early dynamic protein
group and 5 to the late protein group, as shown in the results below.

Accession	Gene Name	cluster	known SGs	aggregator(PMID: 33022891)
P09874	PARP1	early	+	+
P11387	TOP1	early	+	+
P17844	DDX5	early	+	+
P31942	HNRNPH3	early	+	+
P52272	HNRNPM	early	+	+
P55795	HNRNPH2	early	+	+
Q13148	TARDBP	early	+	+
Q96PK6	RBM14	early	+	+
P43243	MATR3	early		+
Q9BQG0	MYBBP1A	early		+
P00374	DHFR	late	+	+
P52292	KPNA2	late	+	+
Q15233	NONO	late	+	+
Q16762	TST	late		+
Q9Y606	PUS1	late		+

Notably, 11 out of 15 aggregators are already established as known SG proteins, including TDP-43 (TARDBP), which we have verified through immunofluorescence. Our findings revealed that TDP-43 exhibits significant localization within SGs during the early stages of heat shock but becomes diffuse with prolonged heat shock (this

data have been incorporated into Figure 5E-F).

Based on these observations, it is plausible to suggest that the impact of protein aggregation might not emerge as a significant concern in our datasets. The presence of aggregators among our identified dynamic proteins, especially those already recognized as SG components, suggests that their propensity to aggregate might be an inherent aspect of their regular SG-related function, rather than representing a confounding factor in our analysis.

b. Unclear data analysis workflow. Authors have not provided clear information on how input peptide material were normalized, how quantified protein abundance were normalized, and how z-score were calculated. How did authors resolve the variability in input material and TMT labeling efficiency between batches? Authors did note that

protein abundance was normalized to '1,000,000', but what normalization method was
used? (e.g., Linear or quantile?) This needs to be included as supplementary
information. How is z-score calculated? The mean abundance of protein and variation
used for z-score – are these derived from multiple timepoints? If so, proteins whose
abundance varies greatly between timepoints will be penalized in this scoring regime,
which may be of most interest for biological context. Lastly, we were unable to read the
supplementary table provided in the PDF format, as columns of numbers are presented
without corresponding gene names. They should be in tabulated format along with raw

data.

Response 1-b:

We appreciate this reviewer's attention to the detailed data analysis procedures. We
have incorporated a comprehensive workflow to provide a holistic view of our strategy
(depicted below).

We have also integrated the data analysis procedures suggested by this reviewer into
the revised manuscript.

a. For the processing of raw data, we utilized MaxQuant with the integrated
Andromeda search engine (v.1.5.4.1). In order to address potential batch effects
and combine the control under non-heat shock conditions, we applied the Combat
method (Johnson, W.E. et al., Biostatistics, 2007) from the sva R package. Initially,
each column of replicates was linearly normalized to an equal amount, which
corresponded to the mean value of all columns. Subsequently, Combat was
applied with the HS0 and HS10 time points designated as the early group, HS20
to HS60 as the middle group, and HS120 to Re10 as the late group in each
replicate. The resulting combat-processed data was further normalized using the
TMM normalization method (Robinson, M.D. et al., Genome biology, 2010).

b. To further refine our analysis, we calculated the fold change by taking the maximum
value observed in the HS10 to Re10 time points compared to the HS0 control as
max/Ctrl, as well as the fold change by comparing the HS0 control to the minimum
value observed in the HS10 to Re10 time points as Ctrl/min. Proteins that
exhibited >1.2-fold changes of max/Ctrl or Ctrl/min in both G3BP1 and CAPRIN1
pull-down datasets were termed as dynamic proteins. Furthermore, within the
subset of proteins with fold changes below 1.2, 85 out of 142 were previously
reported as SG constituents, we termed these proteins as invariable proteins.

Dynamic and invariable proteins were selected for subsequent analyses.
Comprehensive details of the data processing have been included in the revision
and are also illustrated below.

c. Furthermore, the CAPRIN1 dataset of dynamic proteins was input into ClustVis for
scaling and clustering analysis. Scaling was applied to the rows using the variance

scaling method, and the resulted z-scores were exported for curve fitting, as shown
in Figure 1F in the revision.

530 d. The enrichment score of SG-Is was determined using a scaled fold change
calculation, which involved comparing the SG-Is values to the maximum value
observed within the HS10 to Re10 time points. This scaling involved the application
of the variance scaling method, yielding a z-score for each protein.

The method section of the manuscript has been updated.

2. Here authors show that increasing total amount of G3BP1 by two-fold was sufficient
to induce SGs without any stress. This is hard to believe and contradictory to other
publications showing that no SGs are detectable in no stress condition, regardless of
the concentration of overexpressed G3BP1 (Figure 1G in Sanders et al., 2020). Many
others have representative microscopy images showing absence of SGs in no stress
condition. Authors need to state this discrepancy and include quantification of SGs
observed in experiment presented in Fig 2D.

Response:

We sincerely appreciate this reviewer for raising this thought-provoking concern. We
acknowledge the observation that this finding may appear contradictory in light of other
publications that show the absence of SGs in a no-stress condition, regardless of the
concentration of overexpressed G3BP1. Many studies indeed present representative
microscopy images illustrating the lack of SGs under no stress conditions.

In response to this valid point, we would like to emphasize that various studies have
indeed demonstrated that the overexpression of pivotal SG factors, including G3BP1,
G3BP2, and TIA-1, can induce the formation of granules independent of stress
conditions (Kedersha et al., 2016; Kedersha et al., 2005; Panas et al., 2019; Tourriere
et al., 2003; Matsuki, H et al., 2012; Natalie Gilks et al., 2004). However, despite these
compelling findings, the precise mechanisms orchestrating the formation of these
stress-independent granules remain incompletely understood. One possible
explanation is that the overexpression of critical SG factors may prompt their self-
association and subsequent aggregation, culminating in the formation of granule-like
structures.

Within our SG-Is+ cell line, we have observed that under normal growth conditions, the
presence of granules was not widespread, as demonstrated below. Approximately 12%
of the cells exhibited SG-Is, a result in line with the mild overexpression of G3BP1-
EGFP.

To further explore this phenomenon, we incorporated this SG-Is⁺ cell line in our study, seeking to compare its components with those of typical SGs. In the revised version, we have included more detailed information about the percentage of SG-Is⁺ cells (~12%) to provide a clearer understanding of this aspect:

“immunoblotting showed that G3BP1 level in these cells is ~1-fold higher than in wild-
type cells, and ~12% of the cells exhibited EGFP-G3BP1-condensates under normal
growth conditions”

3. This manuscript does not provide means to assess the quality of the ‘SG proteome’
defined by their proteomics data. For instance, they fail to show how their identified
proteome compares to other datasets. There are SG proteomes defined through
literature curation (Youn et al., 2019) or using the same method (Jain et al.,) and other
methods (APEX- Marmor-Kollet et al., 2020, Markmiller et al., 2018 and BioID – Youn
et al., 2018). Validation of a few ‘early’ and ‘late’ proteins is not sufficient to assess the
quality of the data.

Response:

We sincerely appreciate this reviewer’s valuable suggestion. In response to this
concern, we compared our dataset with the previously reported list of SG constituents
obtained from the RNA Granule Database 2.0 (Millar SR. et al., 2023), including
physical studies that utilized prey-based analysis of SGs (Vu et al., 2021; Youn et al.,
2018; Jain et al., 2016; Markmiller et al., 2018; Marmot-Kollet et al., 2021; Padron A et
al., 2019) and co-localization data with SGs. Additionally, a dataset of lost, increased,
and decreased proteins in 2x Ash1-G3BP1 from a proximity-labeling study was also
incorporated in the reported SG proteins list (Yang et al., 2020). The previously
reported list of SG constituents extensively containing the datasets mentioned by this
reviewer.

To visually underscore the assessment of the quality of the “SG proteome” through our

proteomics data, we have included a bar graph illustrating the comparison between
our identified dynamic proteins and the previously established SG constituents (Figure
1C, also depicted below). This graph has replaced the original summary table,
providing a more intuitive representation of the data. Specifically, out of the 385
dynamic and invariable proteins identified in our study, 290 (75%) were previously
recognized as SG constituents. Among these, 139 were categorized as early proteins,
while 66 were classified as late proteins.

Figure 1C. Bar graph shows different groups of proteins.

4. Authors need to acknowledge the limitation of G3BP1-centric purification of SG core
using this method. Specifically, it is limited in terms of the coverage of the SG proteome.
Authors should state that the data shows G3BP1 associated proteins in SG-enriched
fraction, rather than presenting this data as a comprehensive 'SG proteome' [e.g., Line
429]. I strongly recommend immunoprecipitating other SG proteins (those validated to
localizes to SGs during heat shock) to improve the coverage and specificity of the
method used.

Response:

We appreciate this reviewer for raising this critical question and providing insightful
suggestions. We recognize that the G3BP1 pull-down method, utilized for SG
purification, has inherent limitations such as a restricted coverage of the SG proteome
and a potential bias towards G3BP1-interacting proteins.

In response to this concern and with the aim of enhancing the comprehensiveness of
our findings, we have performed immunoprecipitation with a second SG marker protein,
CAPRIN1, in the same time course as in the G3BP1 pull-down experiment, and
acquired the TMT-based quantitative proteomics data. Significantly, we conducted a
comparative analysis between the union and intersection sets of the G3BP1 and
CAPRIN1 datasets. Among the proteins showing a >1.2-fold change relative to the
HS0 control, the intersection set comprised a higher proportion (approximately 75%)
of proteins previously reported as SG constituents, whereas the union section

contained only 59% of reported SG proteins, as depicted below.

In order to enhance the specificity of our data, we selected proteins that exhibited a >1.2-fold change over the HS0 control at least once during the time course, as identified in both the G3BP1 and CAPRIN1 pull-down datasets, for subsequent analysis, as described in the response to major concern 1-a.1.

Furthermore, we have taken this aspect into consideration by incorporating an extensive discussion in our manuscript, highlighting the challenges and limitations associated with the G3BP1-based approach:

“Exploring SG constituents through the pull-down of specific SG markers has inherent limitations, such as a restricted coverage of the SG proteomics, a potential bias towards G3BP1-interacting proteins, and possible disruption of SG integrity. Nevertheless, in our study, the temporal proteomics analysis of SGs under prolonged heat shock-induced conditions was conducted using endogenously expressed G3BP1. This approach eliminates the potential artifacts associated with the use of overexpressed genes during SG purification. Importantly, we performed endogenous SG purification using two SG markers (G3BP1 and CAPRIN1) separately, and only retained those co-identified in both pull-down experiments, which largely enhanced the specificity of our reported SG components.”

5. ‘Late’ proteins include components of translational machineries [translation initiation factors (EIF4A1, EIF2S1) and eIF3 complex components], which are expected to be invariable or recruited early, given that SGs arise from translational machineries. EIF3A in Fig 4B clearly shows that it resides in SGs at the earliest timepoint. To propose that these translational factors as ‘late’ proteins, authors need to correct their conclusion and include validation of additional proteins in this complex.

Response:

We sincerely appreciate the insightful suggestion from this reviewer. We acknowledge the well-established concept that SGs arise from the translational machinery of the cell. Notably, a study by Mateju et al. (Cell, 2020) demonstrated that mRNA translation within SGs was initiated after a prolonged stress period (1 h of sodium arsenite stress). This particular observation lends credence to the possibility that translation initiation factors, among them the eIF3 complex, might not be exclusively designated as early

proteins during SG formation.

In our newly obtained datasets, we have observed that 6 out of 9 eIF3 proteins were
classified as late dynamic proteins, as indicated in Table S1 and shown in the
screenshot below.

Accession	Gene Name	known SGs	cluster
B5ME19	EIF3CL	+	late
O00303	EIF3F	+	late
O75821	EIF3G	+	late
P55884	EIF3B	+	late
P60228	EIF3E	+	late
Q14152	EIF3A	+	late
O15371	EIF3D	+	lvariable
Q13347	EIF3I	+	lvariable
Q9Y262	EIF3L	+	lvariable

In Figure 4B of the revision, we have shown that eIF3A was localized within SGs even
during the earliest time points of heat shock, with this localization further intensifying
at late time points. Additionally, the GO process of translation initiation was also
enriched in late proteins, but not in early ones (Figure 3D). This increased localization
of translation initiation factors within SGs might indicate that the potential translation
localized in SGs starts during later stages of the stress response processes. Based on
these findings, it appears that proteins within the eIF3 complex may function as late-
stage constituents within SGs in the context of our study.

6. [Figure 6] Genetic methods to chemical inhibition would be necessary to support the
function of type II NM in SG fusion.

Response:

We appreciate this reviewer's insightful suggestion aimed at strengthening the
conclusion of our study. We sought to examine the effects of NM II knockdown by
utilizing siRNAs targeting MYH9. However, we encountered challenges as the cells
exhibited morphological damage or abnormal aggregation of the GFP signal, as shown
in the results below. These observations are consistent with other studies that suggest
the substantial inhibition of cell proliferation and migration upon knockdown of the
heavy chains of NM II (MYH9 or MYH10) (Meng Chen et al., Cell Death Discov, 2021;
Longyang Liu et al., Adv Sci (Weinh), 2023). This outcome could potentially be
attributed to the essential role that NM II plays in material transport processes. As a
result, the exploration of NM II via knockdown or knockout approaches posed
challenges, given their substantial impact on cell survival and cellular morphology.

We investigated the role of NM II in SG assembly by activating the NM II complex. To achieve this, we transfected cells with a phosphomimetic MYL9 variant containing a T18D/S19D mutation, which mimics the phosphorylated state of MYL9 and activates myosin functions (Jordan R Beach et al., BMC Cell Biol, 2011). Live cell imaging revealed that MYL9-T18D/S19D significantly accelerated SG formation when compared to the wild-type MYL9. These results strongly support the crucial role of NM II in SG formation, which aligns with the inhibitory effects observed with Blebbistatin. We have added this result into Figure 6G and Supplementary Movie S2. Additionally,

the original Figure 6G was moved to Figure S5E.

Figure 6G. Quantification of SG number per cell at different osmotic stress time.

Minor concerns:

1. Please rephrase/ change the interpretation of these conclusions in ‘early dynamic’ SG proteins.
 - a. [Line 71-72, 394-396] The proposed model suggests that the RNA-centric functions of “early dynamic SG proteins” bespeaks a role in RNA-related cellular processes

during the initial stages of SG formation, however, they do not address the possibility
that their recruitment to SGs is purely a consequence of their potent phase separation
tendencies. Please do not assign function based on correlative observation without
supporting data.

b. [Line 193-195] The high degree centrality of early proteins does not necessitate that
they “play essential roles in holding the entire SG network together”. In fact, their high
centrality may be a consequence of their greater IDR content, which might facilitate

more transient (albeit non-essential) interactions with other proteins. Again, cannot
assign function based on network analysis without supporting data.

c. [Line 37] The disruption of SG dynamics has never been shown to directly cause
neurodegenerative diseases; the pathological inclusions mentioned only mimic those
in true disease contexts.

Response:

Thank you for the valuable suggestion. We have revised manuscript accordingly.

a.

“Together, they coordinate a transition from early SGs that are enriched in RNA-centric
functions to late SGs that exhibit enrichment in a diverse range of stress response
related functions, including intracellular protein transport, stress response, and
charging of amino acids onto cognate tRNAs, as evidenced by GO enrichment
analysis.”

“GO process enrichment supported by early proteins, such as regulation of translation,
RNA stability, and RNA localization (Fig. 3D), are highly RNA-centric, indicating that
early SGs could serve as a hub for RNA-related cellular responses.”

b.

“In particular, early proteins, with significantly higher degree centralities compared to
late and invariable proteins (medians: 37.81, 4.59, and 3.49, respectively; Fig. 3A),
might facilitate more transient interactions with other proteins.”

c.

“Disruption of SG dynamics by disease-linked mutations of certain SG constituents,
such as TDP-43 and TIA1, leads to persistent granules, which is associated with
pathological inclusions and neurodegenerative diseases, such as amyotrophic lateral
sclerosis (ALS) and frontotemporal dementia (FTD).”

2. It is not specified if the 1.2-fold change required to categorize a protein as “dynamic”
is measured between consecutive timepoints, or between any two points of the time
course.

Response:

We appreciate this reviewer’s attention to the detailed data analysis procedures. We
determined the fold change by comparing the HS0 control with the maximum and
minimum value observed within the HS10 to Re10 time points for each protein. Please
refer to the previous response to major concern 1-b. We have diligently revised and
incorporated the comprehensive data processing details into our manuscript.

3. [Line 104-105, Fig 1C] It is not specified what sources were relied upon to determine
a protein’s prior annotation as a SG constituent.

Response:

We sincerely appreciate this reviewer's attention to the intricate data analysis
procedures. We updated our previously reported SG constituents list, which was
obtained from the RNA Granule Database 2.0 (Millar SR. et al., 2023), including
physical studies that utilized prey-based analysis of SGs (Vu et al., 2021; Youn et al.,
2018; Jain et al., 2016; Markmiller et al., 2018; Marmot-Kollet et al., 2021; Padron A et
al., 2019) and co-localization data with SGs. Additionally, a dataset of lost, increased,
and decreased proteins in 2x Ash1-G3BP1 from a proximity-labeling study was also
incorporated in the reported SG proteins list (Yang et al., 2020).

We have thoroughly revised and added these details to our manuscript.

SIDE NOTE

It is interesting that late proteins tend to have lower isoelectric point. Could this be
related to the mechanism where brief acidification of the cytosol occurs in various
stress conditions (including heat shock) induces stress response in eukaryotic cells,
as shown in studies by Triandafillou et al., eLife 2020, Weitzel et al., Experimental Cell
Research 1987; Bright and Ellis, Experimental Physiology, 1992; Munder et al., eLife
2016; Kroschwald et al., eLife 2015? If cytosol is acidified during late time points in the
heat shock, this would neutralize proteins with lower isoelectric points, causing them
to phase separate or aggregate.

Response:

We appreciate reviewer's insightful observation regarding the correlation between late
proteins and their lower isoelectric points. This reviewer's comment of a potential link
to the mechanism involving transient cytosolic acidification during stress conditions is
indeed intriguing. It is conceivable that if the cytosol experiences acidification during
the later stages of heat shock, this could lead to the neutralization of proteins with
lower isoelectric points. This in turn might trigger phase separation or aggregation
processes. Further investigations into this potential mechanistic connection could
provide valuable insights into the behavior of late proteins in SG dynamics.

Typographical Corrections

We appreciate this reviewer's attention to the detailed phrasing. We have fixed the
typos identified by this reviewer.

[Line 32] "Therefore" is an inappropriate conjunctive adverb here.

In this context, we are using "Furthermore" instead of "Therefore".

"Furthermore, SG composition varies with cell type, as well as stress type, intensity,
and duration"

[Line 178] A comma should follow the word "Previously".

We have added a comma after "Previously".

"Previously, proximity labeling studies of G3BP1 in the presence and absence of either

oxidative stress in HEK293 cells or HS in U2OS cells revealed that...”
[Line 204] “Constituent” should be changed to either “constituency” or “constituents”.
We have used “constituents” instead of “constituent” here.
“A shift in the **constituents** of SGs can be assumed to reflect changes in their functional
capacity(ies).”
[Line 302 & 305] “PAPR” should be changed to “PARP” (poly-ADP-ribosyl polymerase)
We have fixed the typo.
“the most abundant and active member of the **PARP** protein family and a major target
for a class of anti-cancer therapies, as an early dynamic protein. **Poly(ADP-ribosylation)**
(PARylation) has been reported to regulate the formation of physiological and
pathological SGs. Several **PARPs** have been reported as constituents of oxidative
stress-induced SGs”
[Line 306 & 307] “Nuclear” should be changed to “nucleus”.
We have changed “nuclear” to “nucleus”.
“PARP1 localizes and functions mainly in the **nucleus** but also in the cytoplasm⁷⁹.
Immunofluorescence imaging shows that, although predominantly localized in the
**nucleus** over the HS cycle...”
[Line 309] “Becoming diffused” should be changed to “becoming diffuse” or simply
“diffusing”.
We have changed “becoming diffused” to “becoming diffuse”.
“Its signal remained enriched till 60 min HS before becoming **diffuse** at 120 min HS
(Fig. 5C-D).”
[Line 321] “As well as the gathering of...” (addition)
We have added an article here.
“as well as **the** gathering of SG constituents through both passive diffusion and active
transport.”
[Line 322] “The microtubule network and its associated motors have been reported to
be involved in...” (addition & corrections)
We have changed “has” to “have” and “involve” to “be involved”.
“The microtubule network and **its** associated motors **have** been reported to **be involved**
in the latter process.”
[Line 360-361] “As a result” is an inappropriate prepositional phrase here.
We have revised the sentence as follows.
“**The result showed that** SG fusion **happens** much more frequently in vehicle-treated
cells (46/50) than in blebb-treated cells (14/50; two proportion Z-test; Fig. S5C-D).”
[Line 361] “Happen” should be changed to “happens”.

We have changed “happen” to “happens”.

“The result showed that SG fusion happens much more frequently in vehicle-treated

cells (46/50) than in blebb-treated cells (14/50; two proportion Z-test; Fig. S5C-D).”

[Line 366] “Promote” should be changed to “promotes”.

We have changed “promote” to “promotes”.

“Detailed analysis of osmotic-stress induced SG particles shows that active NM II

enhances the mobility of SGs and promotes the coalescence of smaller granules into

larger SGs.”

[Line 401-402] “recruitment of these modules seems to be independent of each other,

supported by the...” (correction & deletion)

We have revised “seem” to “seems” and deleted “is”.

“Notably, SG recruitment of these modules seems to be independent of each other,”

[Line 426] “the NM II complex” (addition, keeping format with Line 416)

We have substituted “NM II” with “the NM II complex”.

“Nevertheless, further studies are necessary to investigate whether actin filaments

function together with the NM II complex in promoting the formation of SGs.”

We thank this reviewer for the constructive and insightful comments, which have

helped us to substantially improve our manuscript.

**Reviewer #3**

We are grateful for the supportive and constructive feedback provided by this reviewer.
We appreciate the reviewer for recognizing the novelty of our dataset and
acknowledging its potential as a valuable resource to gain insights into the distinct
recruitment kinetics of specific SG proteins. We have addressed the reviewer's
suggestion by providing a detailed description of how SG constituents were defined,
along with data analysis details. Furthermore, we have included an HS0 control group
and an additional CAPRIN1 pull-down dataset. Each of the concerns raised by this
reviewer has been addressed in detail below and has been highlighted in blue.
Changes to the text are highlighted in red.

Stress granules (SGs) are dynamic, phase-separated RNA-protein condensates that
form in cells in response to various cellular insults. In recent years, there have been
several efforts to catalog the protein constituents of stress granules through proteomic
approaches. While it is known that some properties of granules can change over the
course of prolonged stress and recovery, potential time dependent changes in stress
granule make-up have been largely unexplored. In this manuscript, Shuyao Hu et al.
adapt previously used methods to explore dynamic changes in the stress-granule
profile during the cycle of heat shock and recovery. The authors show that while a
substation portion of the stress granule proteome remains constant over time, many
proteins have a dynamic profile, accumulating at either early or late time points in the
heat stress process. Interestingly, the authors provide data to suggest that early and
late SG proteins have distinct properties, are recruited to SGs as functional modules
and may serve unique roles in regulating SGs. This work is quite novel, as little has
been published about how SG profiles might change over time and is therefore likely
to be of significant interest to the field. Furthermore, the authors provide a specific
example in the case of a myosin module regulating stress granule formation which
highlights how their proteomics data set can provide new insight into stress granule
regulation for future studies. The conclusions are generally well supported, with cell
imaging and immunoblot data helping to validate the proteomics. Notably, there is
potentially a significant issue in the way the authors have defined stress granule
constituents based on the proteomics data that would need to be addressed prior to
publication, along with several more minor points listed below. If the authors can
sufficiently address these points in a revised manuscript, this study would likely have
a significant impact on the field of SG biology.

**Major point:**

One major point that needs to be clarified is how the authors define SG constituents
based on their proteomics data, as all the subsequent results are dependent on this
analysis. In the methods section on starting on line 729, the authors state that proteins
were considered SG constituents if they met one of the following conditions in “both
replicate series”: (1) detected in only SG samples, (2) enriched >2 fold in SG versus
total protein samples. Many of the proteins identified as “dynamic SG proteins” in Table

1 show enrichment in only in one of 2 replicates (e.g. AARS, ZNF638, WDR33) or
enrichment in neither replicate (e.g. ANXA11, TUBB4B, NHP2). It is somewhat
confusing that these examples are termed dynamic SG proteins even though they don't
fit the enrichment versus total protein criteria of "SG constituents" as defined for the
invariable SG proteins. As written, it seems that a protein would be counted as a
dynamic SG protein even if it were increased at a given stress time points in both the
total protein fraction and SG fraction, with no enrichment in the SG fraction. However,
such a protein may be better described as being dynamically globally upregulated
during stress, independent of any recruitment to SGs (notably this appears to be the
case for AIMP1 in replicate 2) and thus should not be considered a dynamic SG
constituent. Furthermore, it is unclear if the authors have used sample with G3BP-IP
from unstressed cells in determining a proteins status as an SG constituent. Previous
studies using MS approaches to identify SG constituents (e.g. Ref38, Jain et. al),
enrichment in stressed vs unstressed G3BP IP samples was used as a criteria rather
than enrichment versus total protein level. The authors need to clarify their criteria and
either adjust their list of dynamic SG proteins or make a case as to why they have
included these proteins in a list of SG constituents rather than simply proteins for which
the total level is dynamically regulated during stress.

Response:

We thank the reviewer for the careful consideration of the detailed data analysis
procedures. We apologize for our oversight and the less rigorous data processing.
Recognizing the absence of an HS0 control group that underwent differential
centrifugation prior to G3BP1-IP, we have thoroughly revised both our datasets and
data processing procedures.

We prepared a new set of samples-HS0, HS10, and HS60-by applying the same
differential centrifugation steps prior to G3BP1 pull-down as the original experiment.
These data were integrated into the original G3BP1 pull-down dataset using the
ComBat processing of sva R package (Johnson et al., 2007). Subsequently, we have
acquired a dataset comprising a control group under non-stress condition and a series
of time points with varying heat-shock treatment durations.

Additionally, we have obtained an additional dataset encompassing a time series
spanning from HS0 to Re10, utilizing CAPRIN1 for the pull-down procedure.

In this context, proteins that exhibited >1.2-fold changes over the HS0 control in at
least one time point, in both G3BP1 and CAPRIN1 pull-down datasets, were
designated as dynamic proteins (as shown in Figure 1C and depicted below).
Conversely, proteins that displayed fold changes below 1.2, 85 out of 142 were
previously reported as SG constituents, we termed these proteins as invariable
proteins.

Figure 1C. Bar graph shows different groups of proteins.

Notably, the analysis results of our newly combined dataset, derived from two distinct
SG marker pull-down experiments, strongly align with the conclusions drawn from our
initial dataset. The SG dynamic proteins showed a remarkably dynamic temporal
behavior throughout the heat shock process.

We have extensively revised the specific numbers and descriptions in our manuscript
to reflect the changes in input data. Changes in the text are highlighted in red.

**Minor issues:**

2. The conclusions from the data concerning recruitment to stress-independent G3BP1
granules (SG-Ls) are potentially interesting, but the data presentation and discussion
are somewhat difficult to follow. The authors define the enrichment score as a the
“relationship” of z scores for unstressed versus stress-induced granules. The
calculation of this enrichment score is unclear to this reviewer, such that I’m not sure
what it means for a protein to have an SG-Ls enrichment score of zero. The authors
state that positive scores indicate HS-independent recruitment which I would interpret
to mean the protein has similar levels in SGs and SG-Ls, but without seeing how the
value is calculated, one might also assume that a score of zero might indicate no
difference in levels between SGs and SG-Ls. Perhaps the author’s could present the
analysis as a of enrichment between SGs and SG-Ls that is more immediately
interpretable? Also, the data in Fig. 2E might be easier to interpret as a frequency
distribution histogram rather than a cumulative distribution.

**Response:**

We apologize for not including the comprehensive data analysis information. Here is
an overview of our study’s methodology and procedures. Specifically, the enrichment
score of SG-Is was determined using a scaled fold change calculation, which involved
comparing the SG-Is values to the maximum value observed within the HS10 to Re10
time points. This scaling involved the application of the variance scaling method,
yielding a z-score for each protein. Since the fold change values were centered around
0, a positive score for the scaled fold change indicates a tendency to enriched in SG-
Is, while a negative score indicates a tendency to enriched in heat shock SGs. We
employed the enrichment score to facilitate a straightforward comparison of SG-Is
enrichment, with positive and negative scores reflecting whether the enrichment
occurs or not. We have also replaced the cumulative distribution with a scatter plot

accompanied by a frequency distribution histogram (as indicated in Figure 2E and
depicted below).

We have incorporated this essential data analysis detail into our manuscript
accordingly.

3. The author's show a comparison between SG-Ls and G3BP1-IP samples, but do
not seem to mention any comparison of the stress-independent G3BP1-IP data to the
identified groups of SG constituents. The authors should include some analysis to
show how protein levels in the unstressed G3BP1 Ip compare to levels detected in the
SGs and use this as a factor in determining whether or not a protein should be
considered a bona fide SG constituent, either of the dynamic or invariable
variety.

Response:

We sincerely appreciate the valuable suggestion provided by this reviewer. By
incorporating the necessary control for the G3BP1 pull-down dataset and including the
CAPRIN1 pull-down dataset, we conducted a comparative analysis between the
unstressed control and the SGs, as previously described. Please refer to the response
to the major point.

Through the incorporation of these new datasets and a comprehensive comparison,
we have further elucidated the distinct characteristics of SGs under heat stress
conditions.

4. The authors might consider including an addition simple table with lists of proteins
in each determined category (i.e. invariable, early and late SG proteins). This would
allow readers to more easily browse the make-up of each protein

type.

Response:

We appreciate the reviewer for the valuable suggestion to enhance the readability of
our manuscript. We have included the lists of early and late proteins from our new
datasets in Table S1.

5. The authors identify as late dynamic SG components. Figure 4B shows that eIF3A
does appear to have increased colocalization with G3BP1 at later time points.
Interestingly however, the eIF3A signal is clearly observed in puncta at the earliest time
point, some of which appear to be G3BP1 negative. This observation highlights one
limitation of defining SGs by a single marker (e.g. G3BP1) and might suggest that
eIF3A is actually recruited to SGs prior G3BP1, or instead that eif3A is recruited to SG
early but is also found in non-SG puncta early in stress (e.g. p-bodies). Also the authors
state that “eIF3 subunits are classified as late... Line244”, but the MS data shows that
5 eIF3 subunits are also included in the invariable category. The authors should
discuss these issues more carefully.

Response:

We thank the reviewer for raising this issue.

While the presence of these G3BP1-independent or G3BP1-weakly associated foci
raises intriguing possibilities, we acknowledge the need for further investigation to
better understand their significance. Our study provides initial insights into the
localization of eIF3A in SGs, but additional experiments and analyses are necessary
to gain a deeper understanding of the nature and functional implications of these
observations.

In our updated datasets, we found that 6 out of 9 eIF3 proteins were classified as late
dynamic proteins, as indicated in Table S1 and shown in the screenshot below. To
provide a more rigorous description, we have refined the statement “eIF3 subunits are
classified as late dynamic SG proteins” to “Based on our proteomic profiling results, 6
out of 9 eIF3 subunits are classified as late dynamic proteins.”

Accession	Gene Name	known SGs	cluster
B5ME19	EIF3CL	+	late
O00303	EIF3F	+	late
O75821	EIF3G	+	late
P55884	EIF3B	+	late
P60228	EIF3E	+	late
Q14152	EIF3A	+	late
O15371	EIF3D	+	lvariable
Q13347	EIF3I	+	lvariable
Q9Y262	EIF3L	+	lvariable

6. It might be helpful to show fixed-cell images of cells prior to any HS at the resting
state so readers can better appreciate the HS-induced puncta
formation.

Response:

We thank the reviewer for the great suggestion on improving the readability of our
manuscript. We have added the non-stressed fix-cell images in Figure 4 and 5.

We thank this reviewer for the constructive and insightful comments, which have

helped us to substantially improve our manuscript.

REVIEWERS' COMMENTS

Reviewer #1 (Remarks to the Author):

I thought this paper was well written, clear and the results are validated. I have no issues with this paper for Nature Comms.

Reviewer #2 (Remarks to the Author):

Authors have addressed my major concerns. I'm pleased to see that heat shock time course data was normalized against a newly acquired HS 0 timepoint. New addition of CAPRIN1 IP proteomics data also improved the quality of the dataset presented.

Major comments to the revised manuscript are below:

1. Non-muscle myosin complex II (NM II) was identified as an early stress granule (SG) protein under heat shock. However, authors examined the role of NM II in SG formation during hyperosmotic stress due to technical issues in using NM II inhibitors during heat shock. Several studies have shown that proteins localize to SGs in context-dependent manner (Markmiller et al., Cell 2018, Aulas et al., J. Cell Sci 2017). Can authors provide evidence that NM II also localizes to SGs during hyperosmotic stress to substantiate NM II's role in SG formation during hyperosmotic stress?

2. In response to my original comment 1-a.2, authors have concluded that aggregation of SG proteins during early heat shock is not a concern using TDP-43 as an example. However, in Fig 5E, TDP-43 shows reduced fluorescence intensity in HS120 and HS180 timepoints. I suspect that TDP-43 aggregated, then degraded during prolonged heat stress. Can authors discuss that the 'early' signature of TDP-43 engagement with SGs and reduced engagement in 'late' timepoints can be attributed to turnover of the protein in prolonged stress?

3. Fig 1E clustergram: please show zoomed in view of gene names or show this as a table to display distinct groups of proteins showing different SG association kinetics.

4. Please provide GO enrichment analysis results as a supplementary table.

Minor

1. Please clarify that three replicates were used for proteomics analysis using G3BP1 and CAPRIN1 IP in the main text and methods section.

2. Grammatical errors

1. line 700. 'SG protein was' -> 'SG proteins were'

Reviewer #3 (Remarks to the Author):

Shuyao Hu present here a revised manuscript detailing dynamic changes to the protein make-up of stress granules formed in response to prolonged heat shock. In response to the concerns of this reviewer and others, the authors have provided important control proteomics samples including a sample collected from unstressed cells as well as a second proteomics data set using a CAPRIN1 as an additional bait protein for stress granule enrichment. Additionally the authors have provided much need clarity on experimental details including the process for determination of proteins as bona-fide stress granule constituents and the comparison of SG constituents with stress-independent constituents of spontaneous G3BP1 granules. Furthermore the authors have provided new data demonstrating that TDP43 is a dynamic SG constituent that appears to leave the SG prior to its dissolution, which is likely to be of further interest to the field due to the strong disease relevance of this protein. Together, this changes to the revised manuscript make it more suitable for publication and more likely to have an important impact on the field of stress granule biology.

Response to reviewers' comment:

Reviewer #1:

I thought this paper was well written, clear and the results are validated. I have no issues with this paper for Nature Comms.

Response: we thank this reviewer for his/her positive feedback and strong supports. We are glad that we have addressed all the questions this reviewer raised.

Reviewer #2

We appreciate the reviewer for his/her thorough examination of the revised manuscript and for providing critical and constructive comments. In response to the reviewer's concerns, we have included additional data for the reviewer's reference. Each concern raised has been addressed in detail below, with corresponding changes to the text highlighted in red for clarity.

Authors have addressed my major concerns. I'm pleased to see that heat shock time course data was normalized against a newly acquired HS 0 timepoint. New addition of CAPRIN1 IP proteomics data also improved the quality of the dataset presented.

Major comments to the revised manuscript are below:

1. Non-muscle myosin complex II (NM II) was identified as an early stress granule (SG) protein under heat shock. However, authors examined the role of NM II in SG formation during hyperosmotic stress due to technical issues in using NM II inhibitors during heat shock. Several studies have shown that proteins localize to SGs in context-dependent manner (Markmiller et al., Cell 2018, Aulas et al., J. Cell Sci 2017). Can authors provide evidence that NM II also localize to SGs during hyperosmotic stress to substantiate NM II's role in SG formation during hyperosmotic stress?

Response 1:

We appreciate the insightful suggestion from this reviewer aimed at strengthening the conclusion of our study. We did investigate the co-localization of MYH9 with SGs. Under 20 min of osmotic stress, we observed that some MYH9 signals (green) were localized at the edge of SGs (red) or adjacent to SGs, as demonstrated below. This observation suggests that MYH9 may interact with components of SGs, influencing their organization (shown here only for the reviewer's reference).

2. In response to my original comment 1-a.2, authors have concluded that aggregation of SG proteins during early heat shock is not a concern using TDP-43 as an example. However, in Fig 5E, TDP-43 shows reduced fluorescence intensity in HS120 and HS180 timepoints. I suspect that TDP-43 aggregated, then degraded during prolonged heat stress. Can authors discuss that the 'early' signature of TDP-43 engagement with SGs and reduced engagement in 'late' timepoints can be attributed to turnover of the protein in prolonged stress?

Response 2:

We thank the reviewer for pointing out this issue. In our 20-min heat shock immunofluorescence, TDP-43 was predominantly localized in SGs, with no additional aggregated foci in the cytoplasm. Additionally, we assessed the TDP-43 expression level under prolonged heat shock by WB, and the results show that TDP-43 was not notably degraded, as demonstrated below.

Based on this observation, we conclude that TDP-43 was not degraded during the 180-min heat shock. Please note that the decrease of TDP-43 intensities in immunofluorescence from HS120 to Re10 min was attributed to the reduction in laser intensity. This adjustment was made to prevent signal overexposure in the nucleolus. Furthermore, we estimated the enrichment of TDP-43 in SGs by comparing its signals in SGs with those in the whole cell, and this comparison did not show the diffusion of TDP-43 from SGs at late heat shock time points.

3. Fig 1E clustergram: please show zoomed in view of gene names or show this as a table to display distinct groups of proteins showing different SG association kinetics.

Response:

We appreciate the reviewer for the suggestions. The gene names of clusters in Figure 1E are provided in Supplementary Data 1, and the raw profiling data can be found in a Source Data file.

4. Please provide GO enrichment analysis results as a supplementary table.

Response:

We appreciate this reviewer's attention to the data availability. The GO results have been included in Supplementary Data 3.

Minor

1. Please clarify that three replicates were used for proteomics analysis using G3BP1 and CAPRIN1 IP in the main text and methods section.

Response:

We thank the review for pointing this out. We conducted three biological replicates for each proteomics analysis, utilizing either G3BP1 or CAPRIN 1 for immunoprecipitation (IP). In response to the review, we have incorporated this clarification into the main text and methods section.

In the main text, line 99-100 was revised as follows:

Subsequently, peptides were TMT-labeled and analyzed by quantitative LC-MS/MS, with each condition comprising three replicates (Fig. 1b and Supplementary Fig. 1b).

In the methods, line 565-566 was included as follows:

TMT-quantitative mass spectrometry was performed in triplicate for both G3BP1 and CAPRIN pull-down experiments.

2. Grammatical errors

line 700. 'SG protein was' -> 'SG proteins were'

Response:

We have changed "SG protein was" to "SG proteins were".

"SG proteins were subsequently isolated using..."

Reviewer #3

Shuyao Hu present here a revised manuscript detailing dynamic changes to the protein make-up of stress granules formed in response to prolonged heat shock. In response to the concerns of this reviewer and others, the authors have provided important control proteomics samples including a sample collected from unstressed cells as well as a second proteomics data set using a CAPRIN1 as an additional bait protein for stress granule enrichment. Additionally the authors have provided much need clarity on experimental details including the process for determination of proteins as bona-fide stress granule constituents and the comparison of SG constituents with stress-independent constituents of spontaneous G3BP1 granules. Furthermore the authors have provided new data demonstrating that TDP43 is a dynamic SG constituent that appears to leave the SG prior to its dissolution, which is likely to be of further interest to the field due to the strong disease relevance of this protein. Together, this changes to the revised manuscript make it more suitable for publication and more likely to have an important impact on the field of stress granule biology.

Response: We appreciate the positive feedback and strong support from this reviewer. We are pleased to note that we have successfully addressed all the raised questions from this reviewer.